# SITUATEDTHINKER: GROUNDING LLM REASONING WITH REAL-WORLD THROUGH SITUATED THINKING

## ABSTRACT

Recent advances in large language models (LLMs) demonstrate their impressive reasoning capabilities. However, the reasoning confined to internal parametric space limits LLMs' access to real-time information and understanding of the physical world. To overcome this constraint, we introduce SITUATEDTHINKER, a novel framework that enables LLMs to ground their reasoning in real-world contexts through *situated thinking*, which adaptively combines both internal knowledge and external information with predefined interfaces. By utilizing reinforcement learning, SITUATEDTHINKER incentivizes deliberate reasoning with the real world to acquire information and feedback, allowing LLMs to surpass their knowledge boundaries and enhance reasoning. Experimental results demonstrate significant performance improvements on multi-hop question-answering and mathematical reasoning benchmarks. Furthermore, SITUATEDTHINKER demonstrates strong performance on unseen tasks, such as KBQA, TableQA, and text-based games, showcasing the generalizable real-world grounded reasoning capability.

## 1 INTRODUCTION

Recent advancements in large language models (LLMs) have been largely driven by their emergent reasoning capabilities in solving complex tasks (Ahn et al., 2024; Sun et al., 2023; Huang & Chang, 2023), representing a substantial leap toward artificial general intelligence (AGI). More recently, long-chain-of-thought (long-CoT) reasoning models, such as OpenAI-o1 (OpenAI, 2024), DeepSeek-R1 (DeepSeek-AI et al., 2025), have substantially improved LLM reasoning capabilities by generating a deliberate thinking process, involving extensive exploration and reflection before concluding the final answer (Chen et al., 2025b). These advancements are largely credited to reinforcement learning (RL) frameworks (Shao et al., 2024; Schulman et al., 2017), which incentivize LLMs to freely explore the reasoning steps solely given a final reward. This is positioned as a pathway to self-evolving LLMs with test-time scaling in reasoning (Muennighoff et al., 2025), potentially advancing the development of stronger intelligence (Snell et al., 2024).

Despite their success, current long-CoT reasoning remains confined to the internal parametric space of LLMs, limiting alignment with the external world. This closed-world reasoning restricts LLMs from accessing up-to-date information and adapting to the ever-evolving world, often leading to hallucinations and factual inconsistencies (Wu et al., 2024; Araya, 2025). Moreover, the absence of an internal world model impairs the ability to reason about physical dynamics (Wang et al., 2023), reducing performance on tasks requiring real-world understanding, such as path planning (Song et al., 2023) and robot control (Singh et al., 2023). These limitations pose significant obstacles to the goal of achieving AGI (Feng et al., 2024), underscoring the necessity for LLMs to interact with and ground their reasoning in the external world.

Existing attempts for aligning LLMs with the external world have primarily focused on using retrieval-augmented generation (RAG) (Wu et al., 2024) or tool-calling (Schick et al., 2023b) to inject external knowledge into LLM reasoning. While enhancing factual accuracy, they raise a fundamental challenge in determining the boundary between the LLMs' internal knowledge and externally retrieved information (Ren et al., 2025) (**C.1**). Over-reliance on either internal knowledge or external information may lead to brittle or suboptimal reasoning. Additionally, complex reasoning tasks often require deliberate, multi-step thinking processes (**C.2**). This necessitates LLMs to adaptively engage with the external world—querying, receiving feedback, incorporating new information, refining their

thinking through reflection and self-correction, instead of relying on a predefined workflow (Trivedi et al., 2023b). Moreover, the dynamic nature of the external world necessitates that LLMs adjust their thinking processes in response to evolving environments (**C.3**). This requires LLMs to develop generalizable real-world grounded reasoning capabilities rather than focusing solely on a specific task and tool, such as searching (Jin et al., 2025; Song et al., 2025; Jin et al., 2025).

To address these challenges, we propose a novel framework, SITUATEDTHINKER, which effectively grounds LLM reasoning with real-world contexts. Extending the internal reasoning of LLMs, we introduce a new paradigm of *situated thinking*, which allows LLMs to adaptively engage with external environments through predefined interfaces. These interfaces provide a unified description of the external world, such as knowledge, tools, and physics environment, allowing LLMs to access real-world information and feedback, facilitating a more accurate and context-aware thinking process. Situated thinking synergizes the internal reasoning of LLMs with situated reasoning in the external world, allowing LLMs to identify required information to surpass its knowledge boundaries, refine their thinking processes, and improve their overall performance in real-world tasks (to address **C.1**).

To facilitate deliberate situated thinking, we adopt the RL framework (Shao et al., 2024) to enable LLMs to reason with the external world and tackle complex tasks (to address **C.2**). During training, LLMs are prompted to use interfaces to gather necessary real-world information and explore their reasoning steps freely to reach a final conclusion. The model is then optimized for the accuracy of its conclusions using a straightforward rule-based reward function, avoiding complex intermediate supervision. To incentivize the generalizable situated thinking capability of LLMs, we train the model on two representative tasks (e.g., multi-hop QA and mathematical reasoning) and two fundamental interfaces (e.g., knowledge retrieval and code execution) to improve its adaptability to out-of-domain external worlds (addressing **C.3**). In general, the distinguishing characteristics of our work are as follows: ❶ SITUATEDTHINKER integrates internal and external knowledge within single-turn thinking to effectively tackle real-world application tasks; ❷ SITUATEDTHINKER is not restricted to single external interfaces, such as code execution or web search, and demonstrates generalizability across tasks in diverse scenarios. Experimental results indicate that various LLMs trained with our SITUATEDTHINKER framework demonstrate significant performance improvements on multi-hop question-answering and mathematical reasoning benchmarks, outperforming prominent baselines. Furthermore, we evaluate the generalization capabilities of SITUATEDTHINKER across unseen tasks, including KBQA, TableQA, and text-based games, demonstrating strong situated thinking capability with new interfaces without further training. Moreover, the empirical analysis reveals that SITUATEDTHINKER can effectively perceive the boundaries of its own knowledge and conduct complex reasoning by appropriately invoking interfaces to reflect and verify uncertain thinking processes. The contributions of this work can be summarized as follows:

- We propose a novel framework, SITUATEDTHINKER, that enables LLMs to ground their reasoning in the external world to extend the capability of LLMs for real-world tasks.

- We introduce situated thinking, a new paradigm that allows LLMs to adaptively engage with external environments and incentivize deliberate reasoning processes with reinforcement learning.

- We conduct extensive experiments demonstrating the effectiveness of SITUATEDTHINKER and its generalizable situated thinking to new tasks and interfaces.

## 2 APPROACH

Figure 1 illustrates the overall framework of SITUATEDTHINKER. In this section, we first present the details of situated thinking, which is central to SITUATEDTHINKER's ability to ground LLM reasoning in the external world by defining interfaces, internal action, and situated action. Next, we describe SITUATEDTHINKER's training process, which encourages LLMs to perform complex reasoning about the real world through a deliberative situated thinking approach.

### 2.1 SITUATED THINKING

The situated thinking is designed to enable LLMs to conduct complex reasoning by combining both internal knowledge and external information, which contains three key components: *interfaces*, *internal action*, and *situated action*.

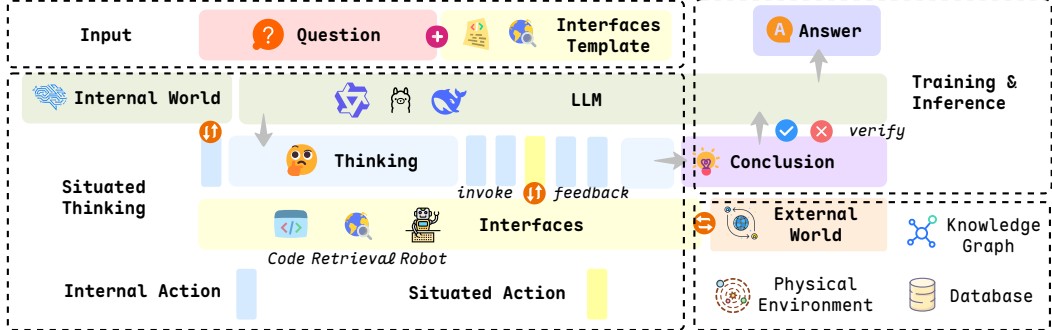

Figure 1: The framework of SITUATEDTHINKER, where LLMs take questions and predefined interfaces as inputs. Then, they conduct situated thinking to adaptively combine basic reasoning with internal action and external reasoning while performing situated actions through the interfaces. The final conclusion is obtained through a deliberate reasoning process and verified to optimize models with reinforcement learning. External world can be presented as knowledge graphs, databases, or the physical environment (like a room space for robot control).

**Interfaces.** Interfaces offer a standardized representation of the external world. We can easily define interfaces for various external environments, such as knowledge graphs, databases, and physical environments to ground LLMs' reasoning in the real world. In SITUATEDTHINKER, we have crafted a universal template for various interfaces, as outlined in the box below. Specifically, each interface is characterized by its `Name` and `Description`, which define the purpose, inputs, and feedback associated with the interface to aid the model in understanding the external world and utilizing the interface effectively. Then, the `Query Format` specifies the format in which the model can interact with the interface, including the start `<interface_start_tag>` and end `<interface_end_tag>` tags for the query. Finally, we assign an `Invoke Limit` to enhance interaction efficiency and prevent the model from entering inefficient interaction loops. Example interfaces like retrieval, code execution, and game control can be found in § B.5.

---

**Interface Template**

**Interface For** {*Interface Name*}
- **Description:** {*Description for the Interface*}
- **Query Format:** `<interface_start_tag>` ...query... `<interface_end_tag>`.
- **Invoke Limit** {*Invoke limit*}.

---

**Internal Action.** The internal action is a fundamental step of the thinking process, enabling step-by-step reasoning though token generation (CoT) (Wei et al., 2022). It highlights the LLMs' ability to perform basic reasoning using their internal knowledge, such as problem decomposition, summarization, and simple arithmetic operations. For example, when presented with the question, *What government position was held by the woman who portrayed Corliss Archer in the film Kiss and Tell?*, LLMs could break down the query into two sub-questions using internal action: 1) *Who was the actress that portrayed Corliss Archer in "Kiss and Tell"?*; and 2) *What government position did Shirley Temple hold?*. Additionally, for some mathematical problems, advanced LLMs can utilize its internal knowledge to perform basic arithmetic operations. For instance, when asked *What is 128 + 56?*, LLMs can internally compute the answer as 184 without needing to invoke any external interface.

**Situated Action.** When addressing tasks requiring up-to-date knowledge, perception of the external environment, or complex reasoning beyond LLMs' capabilities, LLMs must engage with the external world to conduct reasoning, which is called situating action. For example, as shown in § 3.6, the LLM first uses internal actions to reason and analyze questions. It then realizes it lacks information about the current president of East Timor and formulates a query to expand its knowledge by asking: *Who is the current president of East Timor?* through the interface. The query is enclosed within the tags `<interface_start_tag>` and `<interface_end_tag>` of the relevant interface. Then, we invoke the interface to execute the query and conduct the reasoning on the external world to receive feedback that *The current president of East Timor is Francisco Guterres*. The feedback is

returned with the format of `<result>` and `</result>`, which is incorporated into the thinking process to facilitate further reasoning. Additionally, cases in § C.3 show that LLMs could invoke a coding interface to solve complex mathematical reasoning or obtain real-world knowledge.

## 2.2 INCENTIVIZING REASONING WITH SITUATED THINKING USING REINFORCEMENT LEARNING

Complex tasks often require LLMs to conduct deliberate reasoning, utilizing situated thinking to surpass knowledge boundaries while reflecting on feedback from the external world. Teaching LLMs to reason based on real-world information is essential but challenging due to the scarcity of human-annotated data. Reinforcement learning (RL) has emerged as a powerful method for enhancing LLMs' reasoning capabilities by providing rewards based on final conclusions (DeepSeek-AI et al., 2025). Therefore, we aim to harness RL to enhance the reasoning abilities of LLMs, enabling them to explore the external world with situated thinking and incorporate feedback for refining their reasoning and maximizing answer accuracy.

### 2.2.1 INPUT TEMPLATE

The input to SITUATEDTHINKER consists of two main components: the system prompt and the user question.

**System Prompt.** The system prompt is designed to guide LLMs in reasoning and interacting with the external world. We first prompt LLMs to perform a thorough analysis of the problem through a reasoning process that leads to a conclusion, marked by the tags `<conclusion>` and `</conclusion>`. The final answer is clearly presented in the format of `\boxed{...final answer...}`. Then, we provide LLMs with a set of interfaces that allow them to interact with the external world, as detailed in the § 2.1.

---

**System Prompt of SITUATEDTHINKER**

**Reasoning and Format Prompt**
A conversation between a User and an Assistant. The User poses a question, and the Assistant provides a solution. The Assistant's response follows these structured steps:

1. **Reasoning Process**: The Assistant comprehensively thinks about the problem through a reasoning process.
2. **Conclusion**: The Assistant reaches a conclusion, which is enclosed within `<conclusion>` and `</conclusion>` tags. The final answer is highlighted within '\boxed{...final answer...}'.
3. **Response Format**: The complete response should be formatted as:
...reasoning process...
`<conclusion>`
...conclusion...
The answer is \boxed{...final answer...}
`</conclusion>`

**Interfaces Prompt**
During the reasoning process, the Assistant can interact with the system by invoking given interfaces and placing queries within `<interface_start_tag>` ...query here... `</interface_end_tag>` tags. The system processes these queries and returns results in the format `<result>` ...results... `</result>`. After gathering all necessary information, the Assistant continues with the reasoning process to finalize the answer. The assistant cannot invoke each interface more than {`Invoke Limit`} times.
The following are the interfaces provided for the Assistant:

{*Placeholder for Interface Definitions*}

---

**Question.** The system prompt is followed by the specific question, to which the model responds through an iterative and detailed reasoning process of exploration and reflection, accompanied by interaction with the external world.

### 2.2.2 ROLLOUT WITH SITUATED THINKING

The rollout process of SITUATEDTHINKER is designed to enable LLMs to use situated thinking and freely explore the reasoning on the external world. The rollout would generate an iterative reasoning trajectory with both internal and situated actions as detailed in § 2.1. Given a question, we sample $G$ individual reasoning trajectories $\{t_i\}_{i=1}^{G}$ from the policy of the current LLM, denoted as $\pi_{\theta_{old}}(\cdot|q)$ where $q$ is the input question. The trajectories would be assessed by the reward function to optimize the model.

### 2.2.3 REWARD DESIGN

We design a simple reward function to obtain rewards for the generated trajectories. The reward function is based on the format correctness and the answer accuracy of the generated trajectory, which is a common practice in RL training for reasoning tasks (DeepSeek-AI et al., 2025). Formally, the reward $r_i$ for each trajectory $t_i$ is calculated as follows:

$$r_i = \begin{cases} 1.0, & c_{\text{answer}}(t_i), \\ 0.0, & c_{\text{format}}(t_i) \text{ and } \neg c_{\text{answer}}(t_i), \\ -0.1, & \neg c_{\text{format}}(t_i) \text{ and } \neg c_{\text{answer}}(t_i), \end{cases} \tag{1}$$

where $c_{\text{format}}(\cdot)$ evaluates the correctness of the trajectory format, requiring that the conclusion be correctly enclosed within `<conclusion>` and `</conclusion>` tags, and the answer within `\boxed{}`. The $c_{\text{answer}}(\cdot)$ indicates the correctness of the final answer, which will be evaluated by QA accuracy or match correctness. It is noteworthy that we did not design additional rewards for teaching LLMs how to invoke interfaces (Song et al., 2025). In subsequent experiments, we empirically find that the model learned to invoke the interface correctly solely through the reward of answer correctness verification.

### 2.2.4 TRAINING OBJECTIVE

We design the training objective by extending the Group Relative Policy Optimization (GRPO) (Shao et al., 2024). By sampling a group of trajectories, we compute advantages $a_{i,:}$ of all tokens in each trajectory as the mean normalization of group-level rewards $\{r_i\}_{i=1}^{G}$, which is computed as:

$$a_{i,j} = r_i - \text{mean}\left(\{r_i\}_{i=1}^{G}\right), \quad 0 \le j < |t_i|, \tag{2}$$

where $a_{i,j}$ is the advantage of the $j$-th token in the $i$-th trajectory. Then, the final objective of SITUATEDTHINKER is formulated as:

$$\mathcal{L}(\theta) = \mathbb{E}_{(q,a) \sim \mathcal{D}, \{t_i\}_{i=1}^{G} \sim \pi_{\theta_{\text{old}}}(\cdot|q)}$$

$$\left[ \frac{1}{G} \sum_{i=1}^{G} \frac{1}{|t_i|} \sum_{j=1}^{|t_i|} \min\left( \frac{\pi_\theta(t_{i,j}|q, t_{i,<j})}{\pi_{\theta_{\text{old}}}(t_{i,j}|q, t_{i,<j})} a_{i,j}, \text{clip}\left( \frac{\pi_\theta(t_{i,j}|q, t_{i,<j})}{\pi_{\theta_{\text{old}}}(t_{i,j}|q, t_{i,<j})}, 1 - \epsilon_{\min}, 1 + \epsilon_{\max} \right) a_{i,j} \right) \right]. \tag{3}$$

Compared to standard GRPO, our objective incorporates the following key modifications: 1) We introduce distinct clipping bounds, $\epsilon_{\min}$ and $\epsilon_{\max}$, to promote exploration (Yu et al., 2025); 2) We omit the KL penalty term, as our goal is to inject the situated thinking capability into LLM reasoning that is distinct from the base LLMs (Yu et al., 2025).

## 3 EXPERIMENT

### 3.1 EXPERIMENT SETTINGS

**Implementation Details.** We select two distinct base LLMs from the Qwen3 series (Team, 2025): the 8B-Base and 14B-Base models, which have not undergone any post-training, allowing us to observe fundamental performance changes during training. For training, we utilize only two tasks: multi-hop question-answering and mathematical reasoning. Specifically, we employ the training split of MuSiQue (Trivedi et al., 2022) and select 10,000 samples from Big-Math (Albalak et al., 2025) to construct our training data. During the training process, we provide two interfaces for the model: 1) *information retrieval interface*, which retrieves useful information from Wikipedia (2018 dump (Karpukhin et al., 2020)); 2) *code execution interface*, which executes Python code generated by LLMs and returns feedback. More details of interface definitions and training parameters are provided in § B.

**Evaluation Settings.** We evaluate SITUATEDTHINKER on two groups of benchmarks: in-domain and out-of-domain benchmarks. For in-domain benchmarks, we evaluate the performance on four multi-hop question-answering (§ 3.2) and three mathematical reasoning benchmarks (§ 3.3) to assess its reasoning capabilities on trained tasks and interfaces. For out-of-domain benchmarks, we evaluate the generalization capabilities of SITUATEDTHINKER on five unseen external environments, including new domains (e.g., medical, science), new tasks (e.g., KBQA, table QA), and new interfaces (e.g., game environment interaction interfaces) (§ 3.4).

Table 1: Multi-Hop QA benchmarks results. All methods are based on Qwen series LLMs. ◨ indicates the model with 7B/8B parameters and ▬ indicates the model with 14B parameters. **Green** cells indicate the best performance in each column, while **Blue** cells indicate the second-best performance. The results of ReSearch and Search-R1 are borrowed from original papers.

| Model | Size | HotpotQA | 2WikiMultihop QA | MusiQue | Bambooogle |
|---|---|---|---|---|---|
| w/o RAG | ◨ | 0.237 | 0.294 | 0.078 | 0.137 |
|  | ▬ | 0.256 | 0.299 | 0.089 | 0.154 |
| Naive RAG | ◨ | 0.331 | 0.258 | 0.090 | 0.164 |
|  | ▬ | 0.358 | 0.261 | 0.118 | 0.185 |
| Iter-RetGen | ◨ | 0.370 | 0.312 | 0.110 | 0.242 |
| IRCOT | ◨ | 0.340 | 0.246 | 0.107 | 0.282 |
| ReSearch | ◨ | 0.406 | 0.447 | 0.217 | 0.432 |
| Search-R1 | ◨ | 0.380 | 0.326 | 0.168 | 0.384 |
| SITUATEDTHINKER | ◨ | 0.433 | 0.464 | 0.235 | 0.552 |
|  | ▬ | 0.451 | 0.483 | 0.255 | 0.512 |

## 3.2 PERFORMANCE ON MULTI-HOP QUESTION-ANSWERING BENCHMARKS

**Benchmarks.** We first evaluate SITUATEDTHINKER on test the split of four multi-hop question-answering benchmarks: HotpotQA (Yang et al., 2018), 2WikiMultihopQA (Ho et al., 2020), MusiQue (Trivedi et al., 2022), and Bamboogle (Press et al., 2023), with the information retrieval and coding interface seen during training. We report exact match (EM) metrics to assess the performance on these benchmarks.

**Baselines.** We employ four types of baseline methods: 1) No RAG methods, where the LLM is prompted to generate answers directly; 2) Naive RAG methods, involving a straightforward retrieval-based approach that concatenates the retrieval results with the question before prompting the language model to generate an answer; 3) Multi-step RAG methods, which utilize a multi-step RAG framework during reasoning, including two prominent methods: Iter-RetGen (Shao et al., 2023) and IRCoT (Trivedi et al., 2023a); and 4) Search-enhanced models, which utilize search tools to retrieve information during the LLM reasoning process, including two advanced methods: ReSearch (Chen et al., 2025a) and Search-R1 (Jin et al., 2025), which are also trained from GRPO. All baselines are implemented on base models of the equivalent size and family, and we leverage the same external documents for the retrieval. For methods that are challenging to reproduce, we report the best results achieved with models of the same size as presented in the original papers.

**Performance.** Table 1 summarizes the performance of SITUATEDTHINKER in comparison to baseline methods. Notably, SITUATEDTHINKER consistently surpasses prominent baseline approaches. In particular, our model outperforms those baseline methods that incorporate retrieval in their reasoning process, demonstrating its ability to interact with the external environment using multiple interfaces (e.g., coding) beyond simple retrieval (§ C.3). Additionally, our improvements are evident across both in-domain (MusiQue) and out-of-domain benchmarks (e.g., HotpotQA, 2WikiMultiHopQA, and Bamboogle).

## 3.3 PERFORMANCE ON MATHEMATICAL REASONING BENCHMARKS

**Benchmarks.** We assess the mathematical reasoning capabilities of SITUATEDTHINKER and baseline models using three representative benchmarks: AIME24[1], AIME25[2], and MATH500 (Hendrycks et al., 2021; Lightman et al., 2024). The average accuracy at position 32 (avg@32) is employed as the evaluation metric.

**Baselines.** Three categories of baseline methods are employed: 1) Base models, which are public LLMs without additional training, including Qwen3-Base (Team, 2025) and Qwen2.5-Math-Instruct (Yang et al., 2024); 2) Advanced reasoning LLMs trained via reinforcement learning, which employ reinforcement learning with verifiable rewards but cannot interact with real-world, such as SimpleRL-Zero (Zeng et al., 2025) and Eurus-2 (Cui et al., 2025); 3) Tool-integrated reasoning

---

[1] https://huggingface.co/datasets/AI-MO/aimo-validation-aime
[2] https://huggingface.co/datasets/opencompass/AIME2025

methods, which incorporate code integration into their reasoning process, including Qwen2.5-Math-Instruct-TIR (Yang et al., 2024) and ToRL (Li et al., 2025).

**Performance.** As demonstrated in Table 2, SITUATEDTHINKER achieves a significant improvement over the base model. Furthermore, when compared with other advanced baselines, particularly those derived from math-specific models typically pre-trained on extensive professional corpora, our model also demonstrates competitive performance. In comparison to

Table 2: Mathematical reasoning benchmarks results. *Math-Specific* means whether is based on the math-specific LLM.

| Model | Size | Math-Specific | AIME24 | AIME25 | MATH500 |
|---|---|---|---|---|---|
| Qwen3-Base | ▭ | ✗ | 11.7 | 7.6 | 59.6 |
| | ▬ | ✗ | 10.0 | 10.1 | 70.7 |
| Qwen2.5-Math-It | ▭ | ✓ | 10.0 | 16.7 | 74.8 |
| SimpleRL-Zero | ▭ | ✓ | 33.3 | 6.7 | 77.2 |
| Eurus-2 | ▬ | ✓ | 26.7 | 13.3 | 79.2 |
| Qwen2.5-Math-It-TIR | ▭ | ✓ | 26.7 | 16.7 | 74.8 |
| ToRL | ▭ | ✓ | 43.3 | 30.0 | 82.2 |
| SITUATEDTHINKER | ▭ | ✗ | 27.0 | 22.6 | 84.7 |
| | ▬ | ✗ | 43.0 | 26.5 | 87.1 |

the advanced tool-integrated baseline ToRL, which is trained from the stronger math-specific LLM and more training data, SITUATEDTHINKER outperforms it on the MATH500 dataset and achieves competitive results on AIME24 and AIME25.

## 3.4 PERFORMANCE REGARDING GENERALIZATION TO OUT-OF-DOMAIN WORLDS

In this section, we aim to assess the generalization capability of SITUATEDTHINKER with respect to out-of-domain external environments.

**Benchmarks.** We focus on two types of generalization capabilities: 1) cross-domain and 2) cross-interface. For cross-domain generalization, we utilize the medical reasoning benchmark MedQA (Jin et al., 2020) and the scientific reasoning benchmark GPQA (Rein et al., 2023), which share interfaces with the training data but focus on different disciplines. For cross-interface generalization, we evaluate three benchmarks: the knowledge-based question-answering benchmark WebQSP (Yih et al., 2016), the table question-answering benchmark WTQ (Pasupat & Liang, 2015), and the text-based planning benchmark TextWorld (Côté et al., 2018). Please refer to § B.4 for details and evaluation metrics of these benchmarks.

**Baselines.** We primarily compare with Qwen3-Base models (Team, 2025). To better evaluate the effectiveness of SITUATEDTHINKER, we also compared it with the interface-enhanced baseline. Specifically, we utilize the input template to feed to the Qwen3-Base models, so that they can interact with the external world using interfaces, which can be seen as an implementation of ReAct (Yao et al., 2023),. We also include Research and Search-R1, which are only trained to incorporate with single interface to further assess the generalization capability of SITUATEDTHINKER.

**Performance.** As illustrated in Table 3, SITUATEDTHINKER significantly surpasses the baseline models both with and without interface invocation, demonstrating its enhanced ability to generalize to new environments. This generalization includes both different vertical domains and different interfaces. Particularly for TextWorld, a dataset based on simulated physical environments, where the

Table 3: Generalization performance comparison between SITUATEDTHINKER and baselines on MedQA, GPQA, WebQSP, WTQ, and TextWorld. The **Interfaces** column indicates whether the model can interact with external world through interfaces.

| Model | Size | Interfaces | MedQA | GPQA | WebQSP | WTQ | TextWorld |
|---|---|---|---|---|---|---|---|
| Vanilla LLMs | ▭ | ✗ | 3.9 | 6.1 | 41.7 | 6.5 | 10.0 |
| | ▬ | ✗ | 69.6 | 39.4 | 53.1 | 10.5 | 24.0 |
| ReAct | ▭ | ✓ | 5.7 | 5.6 | 22.1 | 35.7 | 8.0 |
| | ▬ | ✓ | 71.9 | 35.9 | 56.3 | 41.6 | 28.0 |
| ReSearch | ▭ | ✓ | 43.6 | 20.2 | 25.2 | 8.7 | 4.0 |
| Search-R1 | ▭ | ✓ | 21.8 | 15.2 | 15.9 | 5.1 | 2.0 |
| SITUATEDTHINKER | ▭ | ✓ | 58.1 | 25.8 | 66.5 | 68.9 | 42.0 |
| | ▬ | ✓ | 77.9 | 53.0 | 68.7 | 69.7 | 94.0 |

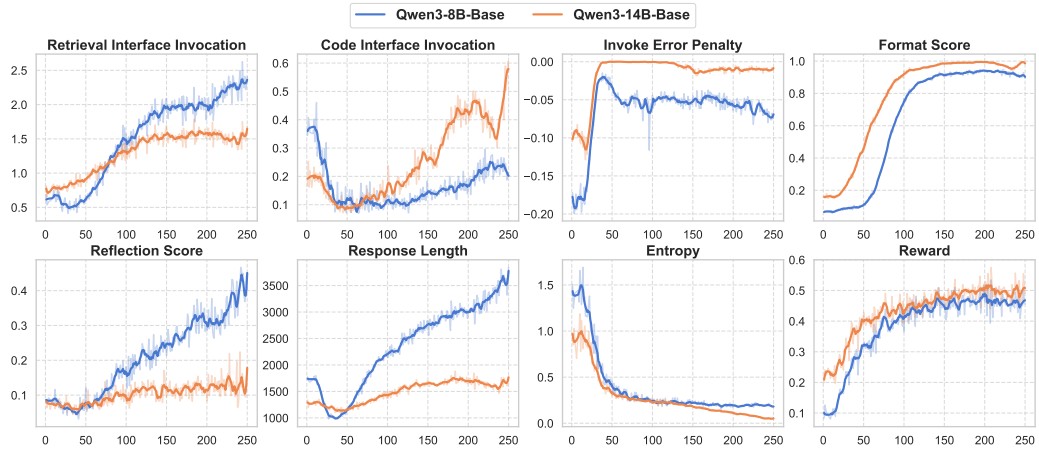

Figure 2: Illustration of training dynamics of SITUATEDTHINKER. The $x$-axis indicates the training steps and the $y$-axis means the observation metrics.

underlying LLMs lack intrinsic knowledge of the environment, effectively utilizing the interface's name to interact with the external world leads to significant performance enhancements.

### 3.5 ANALYSIS ON TRAINING DYNAMICS

In this section, we will explore what the model has learned through GRPO by analyzing the training dynamics. The key training dynamics are detailed in Figure 2.

**SITUATEDTHINKER Learned to Invoke Interfaces.** The training dynamics presented in the sub-figures titled "Code Interface Invocation", "Retrieval Interface Invocation", and "Invoke Error Penalty" illustrate the model's ability to learn correct interface invocation and obtain feedback without relying on explicit reward incentives In the early stages of training, the model is unable to invoke interfaces correctly and demonstrates no significant tendency toward improvement. However, as training progresses, the quantity of successful interface invocations steadily increases, accompanied by a corresponding decrease in error rates.

**SITUATEDTHINKER Learned to Reflect.** The sub-figure titled "Reflection Score" illustrates the dynamics of reflection patterns that emerge in responses generated by SITUATEDTHINKER. As training progresses, the frequency of these reflection patterns gradually increases, indicating that SITUATEDTHINKER is learning to reflect based on external feedback and subsequently improve its reasoning performance. Additionally, the sub-figure titled "Response Length" demonstrates that the length of SITUATEDTHINKER's responses gradually increases as training continues. This aligns with recent research expectations (Snell et al., 2024), suggesting that SITUATEDTHINKER has learned to progressively scale its reasoning by enhancing interface invocation efficiency and incorporating reflection and backtracking to verify the correctness of its thinking.

### 3.6 CASE STUDY

In this section, we will discuss the abilities and nature of SITUATEDTHINKER by analyzing the Case 3.1 in which SITUATEDTHINKER actually performs situated thinking.

**Reasoning Step by Step.** SITUATEDTHINKER firstly decomposes complex reasoning problems and solves them incrementally, a capability acquired automatically during model training without relying on annotated data.

**Discovery Knowledge Boundary.** The **red highlighted** part indicates that SITUATEDTHINKER is capable of recognizing the limitations of its own knowledge. Following GRPO training, SITUATED-THINKER can identify when the knowledge or information necessary to solve a problem surpasses its knowledge boundary, prompting it to invoke situating interfaces to interact with the external environment and obtain feedback that aids its reasoning.

**Reflection with Interfaces.** The **green highlighted** part signifies that SITUATEDTHINKER embodies reflective characteristics within the reasoning process. In response to potential errors in prior reasoning, SITUATEDTHINKER reassesses and deliberates to verify the accuracy of its conclusions. Notably, unlike previous long-chain-of-thought reasoning LLMs, such as DeepSeek-R1, SITUATEDTHINKER integrates situating interfaces during reflection to enhance alignment with external environmental information.

---

**Case 3.1: Case From MusiQue**

QUESTION

Who is the president of the newly declared independent country that has a Commission of Truth and Friendship with the country where Ambelau is located?

RESPONSE

**Alright, I need to find out who is the president of the newly declared independent country that has a Commission of Truth and Friendship with the country where Ambelau is located. Let's break this down step by step.**
### Step 1: Identify the country where Ambelau is located.
...
From the information, it seems that Taur Matan Ruak was the President of East Timor from 20 May 2012 to 20 May 2017. However, I need to confirm if he is the current president or if there have been any changes since then.
**Wait, the information also mentions that José Ramos-Horta was the President of East Timor from 20 May 2007 to 20 May 2012. So, the timeline is: - José Ramos-Horta: 2007-2012 - Taur Matan Ruak: 2012-2017**
**But the question is about the newly declared independent country, which is East Timor, and the president of that country. Since East Timor declared independence in 2002, and the timeline provided goes up to 2017, I need to find out who the current president is.**
**Wait, the information doesn't specify the current president. Maybe I need to check more recent sources.**
`<retrieval>`Who is the current president of East Timor?`</retrieval>`
...
The answer is \boxed{Francisco Guterres}

`</conclusion>`

---

More cases of SITUATEDTHINKER's reasoning process across different tasks and external environments are provided in § C.3.

## 4  RELATED WORK

The most related line of work involves the integration of LLMs with external information through tools or functions which are similar to the interfaces defined in our paper. Previous works (Schick et al., 2023a; Yao et al., 2023; Wang et al., 2024; Gou et al., 2024; Qin et al., 2024; 2025) leverage the capabilities of LLMs' tool-calling, which are typically pre-trained on vast amounts of relevant data and can employ pre-defined tools to solve user tasks across multiple rounds of dialogue. However, most of these methods perform direct reasoning based on external information, they lack the ability to engage in reflective thinking or backtracking and to utilize external information to validate their conclusions during the reasoning process. In contrast, SITUATEDTHINKER integrates external information with single-turn deep thinking, enabling LLMs to engage in reflective thinking through external interfaces during the reasoning process. Recent studies (Chen et al., 2025a; Jin et al., 2025; Song et al., 2025; Li et al., 2025; Feng et al., 2025) have attempted to bolster LLMs with reinforcement learning using web search or writing code, which allow thinking and reflecting with external information. However, training these methods on a single tool limits their ability to generalize to new environments, resulting in inadequate capabilities and inflexible workflow definitions. In contrast, SITUATEDTHINKER achieves interface generalization through a unified interface design, moving beyond reliance on a single interface, and takes the lead in validating this approach across extensive benchmarks. Lastly, advanced reasoning models, such as OpenAI DeepResearch (OpenAI, 2025), are regarded as capable of handling out-of-domain tools. We believe that our work constitutes an important step toward exploring the implementation of such models.

## 5  CONCLUSION

This paper introduces SITUATEDTHINKER, a novel framework that facilitates *situated thinking*, enabling LLMs to actively engage with external environments through predefined interfaces. By employing reinforcement learning, SITUATEDTHINKER promotes intentional reasoning and adaptation to diverse tasks and interfaces. Extensive experiments demonstrate the effectiveness of SITUATEDTHINKER, while further analysis highlights its intriguing and meaningful reasoning behaviors.

## ETHICS STATEMENT

This research focuses solely on general scientific tasks and does not pose risks to health, safety, personal security, or privacy. No human subjects are involved, and no new datasets are released as part of this study. Furthermore, the research does not include potentially harmful insights, methods, or applications, nor does it raise concerns related to privacy, security, legal compliance, or research integrity. Consequently, we anticipate no ethical risks or conflicts of interest. We are committed to maintaining the highest standards of scientific integrity and adhering to ethical guidelines throughout the research process.

## REPRODUCIBILITY STATEMENT

We provide a comprehensive description of the proposed model in the main body, accompanied by detailed implementation specifics, including dataset information, baseline models, and experimental settings in §§ B and 3.1. All datasets used in this research are publicly available. Key code implementations are included in the supplementary materials for reference, with the complete code to be released publicly upon acceptance of the paper.

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

# Appendix

## Table of Contents

# A DISCUSSIONS

**Comparison With RAG.** The classic RAG framework typically operates as a one-off process: it retrieves information based on the initial query and synthesizes a response, treating retrieval and reasoning as separate, disconnected stages. Consequently, when faced with multi-hop reasoning tasks that require connecting disparate pieces of information, traditional methods struggle to capture the latent relationships between multiple sub-questions and necessary intermediate answers. In contrast, our proposed method fundamentally shifts this paradigm by embedding dynamic interaction with the external environment directly into the model's single-turn chain of thought. By harnessing the inherent planning and reasoning strengths of LLMs, the system iteratively solves complex questions step by step—retrieving, evaluating, and reasoning in a loop—thereby achieving a level of accuracy and depth that static retrieval cannot match.

# B MORE IMPLEMENTATION DETAILS

## B.1 TRAINING PARAMETERS

The hyperparameters used for training are listed in Table 4.

Table 4: Training Hyperparameters for SITUATEDTHINKER.

| Parameters | Value |
|---|---|
| Learning Rate | $1e$-6 |
| Total Training Steps | 250 |
| Warmup Steps | 20 |
| # Rollouts per Question | 8 |
| Total Training Batch Size | 256 |
| Max Prompt Length | 2048 |
| Max Response Length | 12288 |
| $\epsilon_{min}$ | 0.2 |
| $\epsilon_{max}$ | 0.28 |

## B.2 DETAILS OF TRAINING DATA

The training data consist of two components: 1) the training subset of MusiQue (Trivedi et al., 2022), containing 19,938 samples; and 2) 10,000 mathematical data from Big-Math (Albalak et al., 2025), where the pass rate in the original data is treated as an indication of difficulty, and the easy (pass rate $\geq 0.7$), medium ($0.3 \leq$ pass rate $< 0.7$), and hard (pass rate $< 0.3$) questions are selected in a ratio of 1:1:8.

## B.3 DETAILS OF HARDWARE AND SOFTWARE

We conduct experiments on the cluster equipped with NVIDIA H100-80G GPUs. The reinforcement learning framework is implemented based on veRL (Sheng et al., 2025), cooperated with Pytorch (Paszke et al., 2019) 2.6.0, Transformers (Wolf et al., 2020) 4.51.3, vLLM (Kwon et al., 2023) 0.8.4.

## B.4 DETAILS OF OUT-OF-DOMAIN BENCHMARKS

**MedQA.** MedQA is a free-form, multiple-choice open-domain question-answering dataset designed to tackle medical problems, as detailed by professional medical licensing exams like the USMLE, AIIMS, and NEET PG (Jin et al., 2020). The dataset encompasses three languages—English, Simplified Chinese, and Traditional Chinese—consisting of 12,723, 34,251, and 14,123 questions, respectively. For our analysis, we utilize only the English test subset, which contains a total of 1,273 questions. Accuracy is reported as the evaluation metric and the information retrieval interface is provided for SITUATEDTHINKER and baselines.

**GPQA.** GPQA (Graduate-Level Google-Proof Q&A) (Rein et al., 2023) is a rigorous multiple-choice benchmark comprising 448 expert-crafted questions in biology, physics, and chemistry. These questions are both authored and validated by domain experts who either hold or are pursuing PhDs in the respective fields. While experts demonstrate an accuracy of only 65%, which increases to 74% when clear mistakes are discounted, skilled non-experts achieve a mere 34% accuracy, despite having full web access and allotting over 30 minutes per question. Our experiment employs the diamond split of GPQA, consisting of 198 questions. Accuracy serves as the evaluation metric and the information retrieval interface is provided for SITUATEDTHINKER and baselines.

**WebQSP.** WebQSP dataset (Yih et al., 2016) is a significant benchmark in knowledge-base question answering, derived from the original WebQuestions dataset, which includes 6,642 question–answer pairs over Freebase. It contains 4,737 fully annotated SPARQL query parses and 1,073 partial annotations for questions that could not be semantically parsed or required descriptive answers, totaling 5,810 annotations. This dataset necessitates models to perform up to two-hop reasoning over Freebase entities, serving as a fundamental benchmark for multi-hop KBQA research. Our experiments are conducted on the test split comprising 1,628 questions. We implement two interfaces for interacting with the knowledge base: 1) the Relation Retrieval Interface for retrieving neighboring relations of a given entity, and 2) the Tail Entity Retrieval Interface for retrieving neighboring tail entities of a given entity and relation. We report hits1, which measures the correctness of the predicted answer, as the evaluation metric.

**WTQ.** WTQ dataset (Pasupat & Liang, 2015) is a large-scale question-answering dataset based on semi-structured HTML tables from Wikipedia, designed for exploring compositional semantic parsing on real-world tables. It includes 22,033 free-form, natural language questions paired with 2,108 distinct tables—each with at least 8 rows and 5 columns—created by Amazon Mechanical Turk workers without templates, resulting in high linguistic and structural diversity. This dataset serves as a benchmark for multi-step reasoning over tables, necessitating operations such as filtering, aggregation, superlatives, arithmetic, joins, and unions. Our experiments are conducted on the test split containing 7,175 questions. We implement three interfaces for interacting with the knowledge base: 1) the Header Interface for retrieving headers given a table ID, 2) the Column Interface for retrieving a column specified by the table ID and header, and 3) the Row Interface for retrieving a row specified by the table ID and row index. We report accuracy as the evaluation metric.

**TextWorld.** TextWorld (Côté et al., 2018) is a text-based game generator and extensible sandbox learning environment for training and testing reinforcement learning (RL) agents. We leverage it to generate a test text-based games benchmark composed of 50 distinct games. We implement four interfaces for interacting with the games through the gym-like APIs(Brockman et al., 2016): 1) the Feedback Interface for returning text observation produced by the game in response to the last command, 2) the Description Interface for returns text description of the current room given the command sequence, 3) the Admissible Commands Interface for returning all commands relevant to the current state given the command sequence, and 4) the Possible Admissible Commands Interface for returning all possible commands of the current game. We report the pass rate of all games as the evaluation metric.

## B.5 DEATILS OF INTERFACE DEFINITIONS

**Question Answering Interfaces.** Two interfaces (e.g., retrieval and code execution) have been employed in HotpotQA, 2WikiMultihop, MuSiQue, Bamboogle, MedQA, and GPQA.

---

**Information Retrieval Interface**

 **Interface For Retrieval Information**
 - **Description:** This interface retrieves the necessary information based on the given query.
 - **Query Format:** ¡retrieval¿ ...query... ¡/retrieval¿.
 - **Invoke Limit** 5.

---

---

**Code Execution Interface**

**Interface For Code Execution**
- **Description:** This interface executes provided Python code snippets and returns the results, making it suitable for tasks such as data processing, analysis, computation, and validation.
- **Query Format:** ¡code¿ ...query... ¡/code¿.
- **Invoke Limit** 5.

---

**Knowledge Graph Interfaces.** We outline the interfaces used in WebQSP to interact with the knowledge graph environments, which includes: relation retrieval and tail entity retrieval interfaces.

**Relation Retrieval Interface**

**Interface For Relation Retrieval**
- **Description:** This interface retrieves the neighboring relations given the entity in the query format ¡relation¿ entity ¡/relation¿.
- **Query Format:** ¡relation¿ ...query... ¡/relation¿.
- **Invoke Limit** 10.

---

**Tail Entity Retrieval Interface**

**Interface For Tail Entity Retrieval**
- **Description:** This interface retrieves the tail entities associated with a given head entity and relation, as specified in the query format ¡entity¿ head entity, relation ¡/entity¿.
- **Query Format:** ¡entity¿ ...query... ¡/entity¿.
- **Invoke Limit** 10.

---

**Database Interfaces.** The interfaces used in WTQ to interact with the database environments include: table header retrieval, column retrieval, and row retrieval interfaces.

**Header Retrieval Interface**

**Interface For Header Retrieval**
- **Description:** This interface retrieves the headers of the table specified by the given table id in the query format ¡header¿ table id ¡/header¿.
- **Query Format:** ¡header¿ ...query... ¡/header¿.
- **Invoke Limit** 10.

---

**Column Retrieval Interface**

**Interface For Column Retrieval**
- **Description:** This interface retrieves a column of the table specified by the given table id and header in the query format ¡column¿ table id, header name ¡/column¿.
- **Query Format:** ¡column¿ ...query... ¡/column¿.
- **Invoke Limit** 10.

---

**Row Retrieval Interface**

**Interface For Row Retrieval**
- **Description:** This interface retrieves a row of the table specified by the given table id and row index in the query format ¡row¿table id, row index¡/row¿.
- **Query Format:** ¡row¿ ...query... ¡/row¿.
- **Invoke Limit** 10.

---

**Game Interaction Interfaces.** In TextWorld, we adopt the interfaces to interact with the game environments, including: obtaining feedback, obtaining description, obtaining admissible commands, obtaining description, and obtaining possible admissible commands interfaces.

---

**Obtaining Feedback Interface**

**Interface For Obtaining Feedback**
- **Description:** This interface returns text observation produced by the game in response to the last command given the game id and the command sequence in the query format ¡feedback¿ game id — command1, command2, ... ¡/feedback¿.
- **Query Format:** ¡feedback¿ ...query... ¡/feedback¿.
- **Invoke Limit** 50.

---

**Obtaining Description Interface**

**Interface For Obtaining Description**
- **Description:** This interface returns text description of the current room given game id and the commmnd sequence in the query format ¡description¿ game id — command1, command2, ... ¡/description¿.
- **Query Format:** ¡description¿ ...query... ¡/description¿.
- **Invoke Limit** 50.

---

**Obtaining Admissible Commands Interface**

**Interface For Obtaining Admissible Commands**
- **Description:** This interface returns all commands relevant to the current state given game id and the command sequence in the query format ¡admissiblecommand¿ game id — command1, command2, ... ¡/dadmissiblecommand¿.
- **Query Format:** ¡admissiblecommand¿ ...query... ¡/admissiblecommand¿.
- **Invoke Limit** 50.

---

**Obtaining Description Interface**

**Interface For Obtaining Description**
- **Description:** This interface returns text description of the current room given game id and the commmand sequence in the query format ¡description¿ game id — command1, command2, ... ¡/description¿.
- **Query Format:** ¡description¿ ...query... ¡/description¿.
- **Invoke Limit** 50.

---

**Obtaining Possible Admissible Commands Interface**

**Interface For Obtaining Possible Admissible Commands**
- **Description:** This interface returns all possible commands given game id in the query format ¡possibleadmissiblecommand¿ game id¡/possibledadmissiblecommand¿.
- **Query Format:** ¡possibleadmissiblecommand¿ ...query... ¡/possibleadmissiblecommand¿.
- **Invoke Limit** 50.

---

## C  MORE EXPERIMENTAL RESULTS

### C.1  CAN INSTRUCTION-TUNED LLMS EFFECTIVELY INVOKE INTERFACES?

In this section, we investigate whether instruction-tuned LLMs can effectively utilize interfaces. We compare the performance of instruction-tuned Qwen3 series models against SITUATEDTHINKER, using a no-thinking mode to minimize the impact of post-training. For instruction-tuned LLMs, we implement a pipeline similar to ReAct (Yao et al., 2023). As shown in Table 5, while instruction tuning enhances the LLM's ability to use interfaces compared to the base model, indicating some improvement in capability, the performance remains inadequate. In contrast, SITUATEDTHINKER significantly outperforms the instruction-tuned models, supporting our claim in **W3** that SITUATED-THINKER can further enhance the capabilities of already powerful models.

Table 5: Comparison of instruct LLMs and SITUATEDTHINKER.

| Model | Size | Musique | Bamboogle | WebQSP | TextWorld |
|-------|------|---------|-----------|--------|-----------|
| Instruct | ▭ | 0.124 | 0.448 | 34.7 | 22.0 |
|  | ▬ | 0.197 | 0.465 | 58.4 | 46.0 |
| SITUATEDTHINKER | ▭ | 0.235 | 0.552 | 66.5 | 42.0 |
|  | ▬ | 0.255 | 0.512 | 68.7 | 94.0 |

Table 6: Ablation study of interfaces during training.

| Training Configuration | Size | Bamboogle | AIME24 | TextWorld |
|------------------------|------|-----------|--------|-----------|
| w/o interfaces | ▭ | 0.208 | 11.3 | 14.0 |
| w/ retrieval only | ▭ | 0.440 | 22.4 | 40.0 |
| w/ code only | ▭ | 0.336 | 24.7 | 36.0 |
| SITUATEDTHINKER | ▭ | 0.552 | 27.0 | 42.0 |

## C.2 ABLATION STUDIES

**Impact of Interfaces in Training.** To investigate the roles of different interfaces in training, we compare the performance of SITUATEDTHINKER when trained without interfaces, with only the Information Retrieval interface, or with only the Code Execution interface, as presented in Table 6. The results indicate that training without interfaces or with a single interface leads to degraded performance. For benchmarks like Bamboogle and TextWorld, which rely heavily on external knowledge, the Information Retrieval interface proves more critical. In contrast, for tasks such as mathematical reasoning (AIME24), where internal knowledge is often sufficient, the Code Execution interface plays a more significant role.

**Impact of Interfaces in Inference.** In this section, we examine the role of different interfaces during the inference phase of SITUATEDTHINKER, with all experiments based on SITUATEDTHINKER trained using both information retrieval and code execution interfaces. First, we assess performance when no interfaces are provided during inference. As shown in Table 7, access to interfaces at inference time significantly enhances performance, particularly for benchmarks like Bamboogle and TextWorld, which heavily depend on external information. Next, we evaluate SITUATEDTHINKER's performance when provided with all 12 interfaces discussed in this paper during inference to determine whether SITUATEDTHINKER can effectively select appropriate interfaces in practical applications and generalize to out-of-domain interfaces, given that interface definitions in real-world scenarios are typically not task-specific. As presented in Table 8, performance degradation is minimal, indicating that SITUATEDTHINKER is robust to a large number of available interfaces and remains focused without being distracted.

## C.3 MORE CASE STUDIES

In this section, we present a detailed analysis of cases sampled from the outputs generated by SITU-ATEDTHINKER across diverse benchmarks to highlight additional aspects of SITUATEDTHINKER's reasoning process.

**Case From Multi-Hop Question-Answering Benchmarks.** Case C.1 illustrates SITUATED-THINKER's response on the Bamboogle dataset. The **red highlighted** and **green highlighted** sections demonstrate SITUATEDTHINKER's ability to correct invocation errors based on external feedback.

**Case From Mathematical Reasoning Benchmarks.** Case C.2 illustrates SITUATEDTHINKER's response on the MATH500 dataset. When addressing fundamental mathematical problems, SITUAT-EDTHINKER can utilize its internal knowledge to perform actions for solving them, as detailed in the **red highlighted** section. The **green highlighted** section indicates that SITUATEDTHINKER leverages the code execution interface to validate its conclusions, enabling effective reflection.

Table 7: Ablation study of no-interfaces during inference.

| Inference Configuration | Size | Bamboogle | AIME24 | TextWorld |
|---|---|---|---|---|
| w/ all interfaces | ▭ | 0.240 | 20.7 | 28.0 |
| SITUATEDTHINKER | ▭ | 0.552 | 27.0 | 42.0 |

Table 8: Ablation study of all-interfaces during inference.

| Inference Configuration | Size | WTQ | TextWorld |
|---|---|---|---|
| w/o interfaces | ▭ | 67.3 | 44.0 |
| SITUATEDTHINKER | ▭ | 68.9 | 42.0 |

**Case From MedQA Benchmark.** Case C.3 presents SITUATEDTHINKER's response on the MedQA dataset, which involves a multiple-choice medical question requiring selection of the correct option. SITUATEDTHINKER demonstrates step-by-step reasoning: it first understands the question and options, then retrieves relevant information (**green highlighted**), and finally draws a conclusion. Before generating the final answer, SITUATEDTHINKER invokes the code execution interface to validate this conclusion (**red highlighted**).

**Case From GPQA Benchmark.** Case C.4 presents SITUATEDTHINKER's response on the GPQA dataset, which includes a multiple-choice science question requiring the correct option to be selected. Initially, SITUATEDTHINKER utilizes its knowledge and computational power to reach a preliminary conclusion (**green highlighted**), followed by invoking a code execution interface to verify this conclusion (**red highlighted**).

**Case from WebQSP Benchmark.** Case C.5 presents SITUATEDTHINKER's response on the WebQSP dataset, involving a question requiring an answer based on knowledge graph data. SITUATEDTHINKER employs step-by-step reasoning: it first understands the question, then identifies and retrieves relevant entity and relation information (**green highlighted**). Before generating the answer, SITUATEDTHINKER verifies its conclusion (**blue highlighted**).

**Case from WTQ Benchmark.** Case C.6 presents SITUATEDTHINKER's response on the WTQ dataset, involving a question requiring an answer based on a table-based database. SITUATEDTHINKER invokes appropriate interfaces to determine the context of the external world (**green highlighted**), subsequently acquires the necessary information from external sources (**red highlighted**) to reach a conclusion, and also utilizes its knowledge to perform simple calculations and reflections.

**Case from TextWorld Benchmark.** Case C.7 presents SITUATEDTHINKER's response on the TextWorld dataset, involving a text-based game requiring a command sequence for completion. The **green highlighted** section shows the initial planning based on understanding the requirements. Subsequently, SITUATEDTHINKER refines its solution by invoking appropriate interfaces to obtain feedback from the game world (**blue highlighted**). Notably, the **red highlighted** section illustrates how SITUATEDTHINKER learns from errors, allowing it to refine its invocation.

---

## Case C.1: Case From Bamboogle

### QUESTION

What is the capital of the second largest state in the US by area?

### ANSWER

austin

### RESPONSE

To determine the capital of the second largest state in the US by area, we need to follow these steps:
1. Identify the second largest state in the US by area. 2. Find the capital of that state.
### Step 1: Identify the Second Largest State in the US by Area
First, we need to know the states ranked by their area. We can use the Code Execution interface to find this information. I'll write a Python code snippet to get the list of US states sorted by area and then identify the second largest state.
¡code¿
```
import requests
from bs4 import BeautifulSoup
url = "https://www.worldatlas.com/articles/the-10-largest-states-in-the-united-states.html"
response = requests.get(url)
soup = BeautifulSoup(response.text, 'html.parser')

# Extract the table of largest states
table = soup.find('table', 'class': 'table')
rows = table.find_all('tr')

# Parse the state names and areas
states = []
for row in rows[1:11]: # Top 10 largest states
cols = row.find_all('td')
state = cols[1].text.strip()
area = float(cols[2].text.strip().replace(',', ''))
states.append((state, area))

# Sort states by area in descending order
states.sort(key=lambda x: x[1], reverse=True)

# Get the second largest state
second_largest_state = states[1][0]
print(second_largest_state)
```
¡/code¿ ¡result¿
Error from code executor: list index out of range
¡/result¿
**It seems there was an error in the code. The error might be due to the structure of the webpage or the way we are trying to parse it. Let's try a different approach to get the information about the largest states by area.**
**Instead of scraping the website, we can use a more reliable dataset. I'll write a Python code snippet with a predefined list of the largest states by area.**
¡code¿
```
# Predefined list of the largest states in the US by area
largest_states_by_area = [ ("Alaska", 1723335.0), ("Texas", 695662.0), ("California", 423967.0), ("Montana", 380831.0), ("New Mexico", 314917.0), ("Arizona", 295234.0), ("Nevada", 286313.0), ("Colorado", 269601.0), ("Oregon", 254755.0), ("Washington", 184827.0) ]

# Sort states by area in descending order
largest_states_by_area.sort(key=lambda x: x[1], reverse=True)

# Get the second largest state
second_largest_state = largest_states_by_area[1][0]
print(second_largest_state)
```
¡/code¿ ¡result¿ Texas
¡/result¿
**The second largest state in the US by area is Texas.**
**### Step 2: Find the Capital of Texas**
**Now that we know the second largest state is Texas, we need to find its capital. We can use the Retrieval interface to get this information.**
¡retrieval¿ What is the capital of Texas? ¡/retrieval¿ ¡result¿ ...retrieval results... ¡/result¿ From the retrieved information, we can conclude that the capital of Texas is Austin.
¡conclusion¿
The answer is \boxed{Austin}
¡/conclusion¿

---

## Case C.2: Case From MATH500

### QUESTION

A regular hexagon can be divided into six equilateral triangles. If the perimeter of one of the triangles is 21 inches, what is the perimeter, in inches, of the regular hexagon?

### ANSWER

42

RESPONSE

To solve the problem, we need to determine the side length of the equilateral triangle first, and then use that information to find the perimeter of the regular hexagon.

**1. Determine the side length of the equilateral triangle:**
**- The perimeter of an equilateral triangle is the sum of the lengths of its three equal sides.**
**- If the perimeter of the equilateral triangle is 21 inches, then the side length $s$ of the triangle can be calculated as:**

$$s = \frac{\text{perimeter}}{3} = \frac{21}{3} = 7 \text{ inches}$$

2. Relate the side length of the triangle to the side length of the hexagon:
- A regular hexagon can be divided into six equilateral triangles, and each side of the hexagon is equal to the side length of these triangles.
- Therefore, the side length of the hexagon is also 7 inches.
3. Calculate the perimeter of the regular hexagon:
- A regular hexagon has six equal sides.
- If the side length of the hexagon is 7 inches, then the perimeter $P$ of the hexagon is:

$$P = 6 \times s = 6 \times 7 = 42 \text{ inches}$$

**Now, let's confirm the calculation using the provided interfaces. ¡code¿**
**s = 21 / 3**
**P = 6 * s**
**P**
**¡/code¿**
**¡result¿ 42.0**
**¡/result¿**
Now we have confirmed the calculation using code execution. The perimeter of the regular hexagon is indeed 42 inches.
¡conclusion¿
The answer is \boxed{42}
¡/conslusion¿

---

## Case C.3: Case From MedQA

QUESTION

A 35-year-old man comes to the physician because of itchy, watery eyes for the past week. He has also been sneezing multiple times a day during this period. He had a similar episode 1 year ago around springtime. He has iron deficiency anemia and ankylosing spondylitis. Current medications include ferrous sulfate, artificial tear drops, and indomethacin. He works as an elementary school teacher. His vital signs are within normal limits. Visual acuity is 20/20 without correction. Physical examination shows bilateral conjunctival injection with watery discharge. The pupils are 3 mm, equal, and reactive to light. Examination of the anterior chamber of the eye is unremarkable. Which of the following is the most appropriate treatment?
A): Erythromycin ointment
B): Ketotifen eye drops
C): Warm compresses
D): Fluorometholone eye drops

ANSWER

B

RESPONSE

Alright, I have this multiple-choice question about a 35-year-old man with itchy, watery eyes and sneezing. I need to figure out the most appropriate treatment from the options provided. Let's break this down step by step.
### Step 1: Understand the Patient's Symptoms and History
First, let's summarize what's given:
- **Patient's Age and Gender:** 35-year-old man.
- **Chief Complaint:** Itchy, watery eyes for the past week, sneezing multiple times a day.
- **History:** Similar episode 1 year ago around springtime.
- **Medical Conditions:** Iron deficiency anemia, ankylosing spondylitis.
- **Current Medications:** Ferrous sulfate, artificial tear drops, indomethacin.
- **Occupation:** Elementary school teacher.
- **Vital Signs:** Within normal limits.
- **Visual Acuity:** 20/20 without correction.
- **Physical Examination:**
- Bilateral conjunctival injection with watery discharge.
- Pupils: 3 mm, equal, reactive to light.
- Anterior chamber of the eye: Unremarkable.
Okay, so the patient has seasonal symptoms of itchy, watery eyes and sneezing, which suggests an allergic reaction. The physical exam shows signs of conjunctival inflammation, which is consistent with allergic conjunctivitis.
### Step 2: Recall the Options
The options for treatment are:
A) Erythromycin ointment
B) Ketotifen eye drops
C) Warm compresses
D) Fluorometholone eye drops
I need to determine which of these is the most appropriate treatment for allergic conjunctivitis.

### Step 3: Retrieve Relevant Information
First, I should retrieve some information about allergic conjunctivitis and the treatments listed.
¡retrieval¿ What is allergic conjunctivitis and what are the common treatments? ¡/retrieval¿ ¡result¿ ...results.. ¡/result¿
From the retrieval, it seems that allergic conjunctivitis is caused by an allergic reaction, often due to hay fever, and is characterized by symptoms like redness, swelling, itching, and watery discharge. The treatment typically involves avoiding allergens and using antihistamines, either topically or systemically.
### Step 4: Analyze Each Option
Now, let's look at each option and see if it aligns with the treatment of allergic conjunctivitis.
**A) Erythromycin ointment**
Erythromycin is an antibiotic. I recall that antibiotics are used to treat bacterial infections, not allergic conditions. So, this doesn't seem appropriate for allergic conjunctivitis.
**B) Ketotifen eye drops**
Ketotifen is an antihistamine. From the retrieval, antihistamines are a common treatment for allergic conjunctivitis. This seems like a good candidate.
**C) Warm compresses**
Warm compresses are often used for conditions like styes or blocked tear ducts. I'm not sure if they're a standard treatment for allergic conjunctivitis. Maybe for symptom relief, but not a primary treatment.
**D) Fluorometholone eye drops**
Fluorometholone is a corticosteroid. Corticosteroids are anti-inflammatory and can be used for severe allergic reactions, but they're usually not the first-line treatment for allergic conjunctivitis due to potential side effects. Antihistamines are generally preferred first.
### Step 5: Narrow Down the Options
Based on the analysis:
- **A) Erythromycin ointment:** Unlikely, as it's an antibiotic for bacterial infections.
- **B) Ketotifen eye drops:** Likely, as it's an antihistamine for allergic conditions.
- **C) Warm compresses:** Possibly for symptom relief, but not a primary treatment.
- **D) Fluorometholone eye drops:** Possible for severe cases, but not the first-line treatment.
So, the most appropriate treatment seems to be **B) Ketotifen eye drops**.
### Step 6: Verify with Code Execution
To ensure I'm correct, I'll run a quick code snippet to confirm the classification of these treatments.
¡code¿
import pandas as pd
# Create a DataFrame with treatments and their categories
data = {
'Treatment': ['Erythromycin ointment', 'Ketotifen eye drops', 'Warm compresses', 'Fluorometholone eye drops'],
'Category': ['Antibiotic', 'Antihistamine', 'Symptom Relief', 'Corticosteroid']
}
df = pd.DataFrame(data)
# Define the correct treatment based on the analysis
correct_treatment = 'Ketotifen eye drops'
# Check the category of the correct treatment
category = df[df['Treatment'] == correct_treatment]['Category'].values[0]
print(f"The correct treatment is {correct_treatment}, which is a {category}.")
¡/code¿ ¡result¿ The correct treatment is Ketotifen eye drops, which is a Antihistamine.
¡/result¿
Yes, Ketotifen is indeed an antihistamine, which aligns with the treatment of allergic conjunctivitis.
### Step 7: Final Answer
After going through the reasoning process and verifying with the code execution, the most appropriate treatment for this patient's allergic conjunctivitis is Ketotifen eye drops.
...
The answer is \boxed{B}.
¡conculsion¿

## Case C.4: Case From GPQA

### QUESTION

Researchers are attempting to detect transits of two Earth-like planets: Planet_1 and Planet_2. They have limited observing time and want to observe the one that has the highest probability of transiting. Both of these planets have already been detected via the RV method, allowing us to know their minimum masses and orbital periods. Although both planets share the same masses, the orbital period of Planet_1 is three times shorter than that of Planet_2. Interestingly, they both have circular orbits. Furthermore, we know the masses and radii of the host stars of these two planets. The star hosting Planet_1 has a mass that is twice that of the host star of Planet_2. As the host of Planet_2 is slightly evolved, both host stars have the same radii. Based on the provided information, the researchers have chosen to observe:
A): Planet_1 is preferred due to its 1.65 times higher probability to transit.
B): Planet_1 is preferred due to its 2.7 times higher probability to transit.
C): Planet_2 is preferred due to its 2.25 times higher probability to transit.
D): Planet_2 is preferred due to its 1.5 times higher probability to transit.

### ANSWER

A

### RESPONSE

1. **Understanding the Problem**: We need to determine which planet has a higher probability of transiting its host star, given the information about their orbital periods and the masses/radii of their host stars. The probability of a transit is related to the inclination of the planet's orbit, and planets with shorter orbital periods around smaller stars have a higher probability of transiting.
2. **Key Factors**:

- **Orbital Period**: Planet_1 has an orbital period that is three times shorter than Planet_2.
- **Host Star Mass**: The host star of Planet_1 is twice as massive as that of Planet_2.
- **Host Star Radius**: Both host stars have the same radius.

3. **Transit Probability**: The probability $P$ of a transit is approximately the ratio of the star's radius $R_*$ to the semi-major axis $a$ of the planet's orbit:

$$P \approx \frac{R_*}{a}$$

For circular orbits, the semi-major axis $a$ can be estimated using Kepler's Third Law:

$$a = \left( \frac{GM_* P^2}{4\pi^2} \right)^{1/3}$$

where $G$ is the gravitational constant, $M_*$ is the mass of the star, and $P$ is the orbital period.

4. **Comparing the Two Planets**:
- Let $P_1$ and $P_2$ be the orbital periods of Planet_1 and Planet_2, respectively. Given $P_1 = \frac{1}{3}P_2$.
- Let $M_{*1} = 2M_{*2}$ and $R_{*1} = R_{*2} = R_*$.

5. **Calculating the Semi-Major Axes**:
- For Planet_1:

$$a_1 = \left( \frac{GM_{*1} P_1^2}{4\pi^2} \right)^{1/3} = \left( \frac{G \cdot 2M_{*2} \cdot \left(\frac{1}{3}P_2\right)^2}{4\pi^2} \right)^{1/3}$$

- For Planet_2:

$$a_2 = \left( \frac{GM_{*2} P_2^2}{4\pi^2} \right)^{1/3}$$

6. **Simplifying the Ratio $\frac{a_1}{a_2}$**:

$$\frac{a_1}{a_2} = \left( \frac{2M_{*2} \cdot \left(\frac{1}{3}P_2\right)^2}{M_{*2} P_2^2} \right)^{1/3} = \left( \frac{2}{9} \right)^{1/3} \approx 0.66$$

So, $a_1 \approx 0.66a_2$.

7. **Calculating the Transit Probabilities**:
- For Planet_1:

$$P_1 \approx \frac{R_*}{a_1} \approx \frac{R_*}{0.66a_2} \approx 1.52\frac{R_*}{a_2}$$

- For Planet_2:

$$P_2 \approx \frac{R_*}{a_2}$$

Therefore, $P_1 \approx 1.52P_2$.

8. **Interpreting the Results**: The transit probability of Planet_1 is approximately 1.52 times higher than that of Planet_2. This means Planet_1 is more likely to transit its host star.

9. **Matching with the Options**: The closest option to a 1.52 times higher probability is option A, which states that Planet_1 is preferred due to its 1.65 times higher probability to transit.

**Let's verify the calculations using Python to ensure accuracy.**
**¡code¿**
```
import math
# Given data
P2 = 1 # Let P2 be 1 unit
P1 = P2 / 3
M2 = 1 # Let M2 be 1 unit
M1 = 2 * M2
# Calculating semi-major axes
a1 = (M1 * P1**2)**(1/3)
a2 = (M2 * P2**2)**(1/3)
# Calculating transit probabilities
P1_transit = 1 / a1
P2_transit = 1 / a2
# Ratio of transit probabilities
P1_P2_ratio = P1_transit / P2_transit
print(P1_P2_ratio)
```
**¡/code¿ ¡result¿ 1.6509636244473134**
**¡/result¿ The calculated ratio of the transit probabilities is approximately 1.65, which matches option A.**
¡conclusion¿
Planet_1 is preferred due to its 1.65 times higher probability to transit.
The answer is $backslash$boxed{A}

¡/conclusion¿

---

### Case C.5: Case From WebQSP

QUESTION

where is jamarcus russell from

ANSWER

mobile

RESPONSE

To find out where JaMarcus Russell is from, I need to follow these steps:

1. First, I should find the neighboring relations of JaMarcus Russell. This will give me an overview of the information available about him.

**Query: ¡relation¿JaMarcus Russell¡/relation¿ ¡result¿ ['sports.sports_team_roster.player', 'common.topic.notable_for', 'education.education.institution', 'american_football.player_rushing_statistics.player', 'sports.pro_sports_played.sport', 'american_football.football_player.position_s', 'sports.pro_sports_played.athlete', 'american_football.player_passing_statistics.player', 'american_football.player_rushing_statistics.season', 'sports.sports_league_draft_pick.school', 'people.person.nationality', 'sports.sports_team_roster.team', 'people.person.parents', 'people.place_lived.person', 'people.place_lived.location', 'people.person.place_of_birth', 'sports.sports_league_draft_pick.draft', 'sports.sports_league_draft_pick.player', 'freebase.valuenotation.has_value', 'education.education.student', 'common.topic.notable_types', 'sports.sports_league_draft_pick.team', 'american_football.player_passing_statistics.team', 'american_football.player_passing_statistics.season', 'american_football.player_game_statistics.team', 'people.person.ethnicity', 'sports.sports_team_roster.position', 'common.topic.image', 'american_football.player_rushing_statistics.team', 'american_football.player_game_statistics.season', 'people.person.profession', 'people.person.gender', 'american_football.player_game_statistics.player'] ¡/result¿**
**The result shows various relations related to JaMarcus Russell. The most relevant one to find his place of birth is 'people.person.place_of_birth'.**

2. Next, I need to retrieve the entity associated with the relation 'people.person.place_of_birth' for JaMarcus Russell.
Query: ¡entity¿JaMarcus Russell, people.person.place_of_birth¡/entity¿ ¡result¿ ['Mobile'] ¡/result¿
The result indicates that the place of birth for JaMarcus Russell is Mobile.

**3. To provide a more complete answer, I should find out more about Mobile, such as the country it is located in. First, I need to find the neighboring relations of Mobile.**
**Query: ¡relation¿Mobile¡/relation¿ ¡result¿ ['music.composition.composer', 'travel.travel_destination.tourist_attractions', 'common.topic.notable_for', 'music.composition.recordings', 'music.recording.producer', 'music.recording.releases', 'music.recording.song', 'location.administrative_division_capital_relationship.administrative_division', 'base.wikipedia_infobox.settlement.area_code', 'music.recording.canonical_version', 'location.location.nearby_airports', 'location.hud_county_place.county', 'film.film_location.featured_in_films', 'freebase.valuenotation.has_value', 'music.single.versions', 'location.location.events', 'music.composition.form', 'common.topic.notable_types', 'location.hud_county_place.place', 'location.citytown.postal_codes', 'music.recording.artist', 'common.webpage.topic', 'government.government_position_held.jurisdiction_of_office', 'location.place_with_neighborhoods.neighborhoods', 'location.location.time_zones', 'common.topic.image', 'location.location.containedby', 'location.statistical_region.population', 'periodicals.newspaper_circulation_area.newspapers', 'location.administrative_division_capital_relationship.capital'] ¡/result¿**
**The result doesn't directly show the country, so I need to find a relation that can lead me to that information. The relation 'location.location.containedby' might help.**

**4. I need to retrieve the entity associated with the relation 'location.location.containedby' for Mobile.**
**Query: ¡entity¿Mobile, location.location.containedby¡/entity¿ ¡result¿ ['Alabama', 'Mobile County', 'United States of America', 'Area code 251'] ¡/result¿**
**The result includes 'United States of America', which is the country. For more specificity, I can check the relation 'location.hud_county_place.county' to find the state.**
**5. I need to retrieve the entity associated with the relation 'location.hud_county_place.county' for Mobile.**
**Query: ¡entity¿Mobile, location.hud_county_place.county¡/entity¿ ¡result¿ ['Mobile County'] ¡/result¿**
**This confirms that Mobile is in Mobile County, Alabama, USA.**
Based on this information, I can conclude that JaMarcus Russell is from Mobile, Alabama, USA.
¡conclusion¿
JaMarcus Russell is from Mobile.
The answer is \boxed{Mobile}

¡/conclusion¿

---

### Case C.6: Case From WTQ

QUESTION

How many were played on a hard surface?

ANSWER

7

RESPONSE

Alright, I have this question: "How many were played on a hard surface?" and I'm given a table with the ID 'nt-458'. I need to figure out how to extract the relevant information from this table to answer the question. Let's break this down step by step.

### Step 1: Understand the Question

**First, I need to understand what the question is asking. It's asking for the number of something that was played on a hard surface. But what exactly is "something"? Is it games, sports, matches, or something else? Since I don't have the actual table, I'll need to figure out the structure of the table to proceed.**

### Step 2: Retrieve the Table Headers

**To understand the table, I should start by retrieving its headers. This will give me an idea of what columns are available and what kind of data is stored in them.**

**I'll use the \*\*Header Retrieval\*\* interface for this.**
**¡header¿nt-458¡/header¿ ¡result¿ ['Outcome', 'Date', 'Tournament', 'Surface', 'Partnering', 'Opponent in the final', 'Score in the final'] ¡/result¿**
**Okay, so the table has the following headers: Outcome, Date, Tournament, Surface, Partnering, Opponent in the final, and Score in the final.**
### Step 3: Identify Relevant Columns
Now, I need to identify which columns are relevant to the question. The question is about something played on a hard surface. Looking at the headers, the "Surface" column seems directly related to the type of surface on which something was played.
So, the "Surface" column is likely the key here. But to get the count of how many were played on a hard surface, I might need to look at the entire "Surface" column and count the occurrences of "hard surface."
**Wait, but the question is a bit vague. It's asking "how many were played on a hard surface?" So, I need to determine what "were played" refers to. Given the headers, it could be matches, games, or some other event. Maybe it's tennis matches, given the presence of "Tournament" and "Score in the final."**
**Assuming it's tennis matches, then "Surface" would indicate the type of court (e.g., hard, clay, grass). So, I need to count how many matches were played on a hard surface.**
### Step 4: Retrieve the "Surface" Column
To get the count, I need to retrieve the "Surface" column from the table.
I'll use the \*\*Column Retrieval\*\* interface for this.
¡column¿nt-458, Surface¡/column¿ ¡result¿ ['Clay', 'Clay', 'Hard', 'Hard', 'Hard', 'Grass', 'Grass', 'Hard', 'Hard', 'Hard', 'Hard']
¡/result¿
Okay, so the "Surface" column contains the following values: ['Clay', 'Clay', 'Hard', 'Hard', 'Hard', 'Grass', 'Grass', 'Hard', 'Hard', 'Hard', 'Hard'].
### Step 5: Count the Occurrences of "Hard"
Now, I need to count how many times "Hard" appears in this list.
Let's go through the list:
1. Clay
2. Clay
3. Hard
4. Hard
5. Hard
6. Grass
7. Grass
8. Hard
9. Hard
10. Hard
11. Hard
Counting the "Hard" entries:
3, 4, 5, 8, 9, 10, 11 → That's 7 times.
**Wait, let's double-check:**
1. Clay
2. Clay
3. Hard
4. Hard
5. Hard
6. Grass
7. Grass
8. Hard
9. Hard
10. Hard
11. Hard
Yes, "Hard" appears 7 times.
### Step 6: Verify the Count
**To ensure accuracy, I'll recount:**
**3, 4, 5, 8, 9, 10, 11 → 7 times.**
Yes, it's correct.
### Step 7: Formulate the Answer
The question is asking for the number of matches played on a hard surface. Based on the data, that number is 7.
¡conclusion¿
The number of matches played on a hard surface is \boxed{7}

¡/conclusion¿

---

### Case C.7: Case From TextWorld

QUESTION

Game Objective:
You are now playing a exciting session of TextWorld! First, it would be a great idea if you could attempt to venture west. Okay, and then, move west. That done, retrieve the staple from the chair inside the office. Once you have got the staple, head east. Then, rest the staple on the shelf inside the cellar. Alright, thanks!

ANSWER

[Any Command Sequence to Win the Game]

RESPONSE

To solve this game, I need to follow the instructions provided in the game objective. Here's the step-by-step reasoning process:
**1. \*\*Start by moving west\*\*: The first instruction is to move west. I should begin with the command 'go west'.**

**2. \*\*Retrieve the staple from the chair inside the office\*\*: After moving west, I need to find the office, get the staple from the chair, and pick it up. The command to pick up an item is usually 'get [item]'. However, I first need to navigate to the office and identify the staple.**
**3. \*\*Head east\*\*: After retrieving the staple, I need to move east. The command for this is 'go east'.**
**4. \*\*Rest the staple on the shelf inside the cellar\*\*: Finally, I need to go to the cellar, find the shelf, and place the staple there. The command to put an item on something is usually 'put [item] on [location]'.**

Let's start by moving west. I'll invoke the \*\*Interface For Obtain Admissible Commands\*\* to see what commands are available initially.

**<admissiblecommand>14</admissiblecommand> <result> Invalid query format. Please use the format <admissiblecommand>game id — command1, command2, ... </dadmissiblecommand>. </result>**

**It seems I made a mistake in the query format. Let's correct that. I'll try again:**

**<admissiblecommand>14 — </admissiblecommand> <result> ['drop burger', 'eat burger', 'examine burger', 'go west', 'inventory', 'look'] </result>**

**Great, the admissible commands initially include 'go west'. I'll execute that command.**

1. \*\*Command\*\*: 'go west'

**Now, I need to check the new state of the game to see where I am and what I can do next. I'll invoke the \*\*Interface For Obtain Description\*\* to get a description of the current room.**

<description>14 — go west</description> <result>
-= Cellar =-
You have come into a cellar. Not the cellar you'd expect. No, this is a cellar.
Look over there! a shelf. You shudder, but continue examining the shelf. The shelf is standard. But oh no! there's nothing on this piece of junk. You make a mental note to not get your hopes up the next time you see a shelf in a room. You can see a counter. The counter is normal. But the thing hasn't got anything on it.
There is a closed hatch leading north. You need an unguarded exit? You should try going east. There is an unblocked exit to the west. There is a fondue on the floor.
</result>

From the description, I'm now in the cellar. The next instruction is to retrieve the staple from the chair inside the office. Since I'm in the cellar, I need to move to the office. The description mentions an unblocked exit to the west, so I'll go west again.

2. \*\*Command\*\*: 'go west'

**I'll check the new room description to see if I'm in the office and to identify the staple.**

<description>14 — go west, go west</description> <result>
-= Office =-
I am required to announce that you are now in the office. You try to gain information on your surroundings by using a technique you call "looking."
You see a chair. The chair is standard. On the chair you see a staple.
There is an unblocked exit to the east.
</result>

Now I'm in the office, and the description mentions a staple on the chair. I'll retrieve the staple.

3. \*\*Command\*\*: 'get staple'

**After getting the staple, I should check the admissible commands to see what I can do next.**

<admissiblecommand>14 — go west, go west, get staple</admissiblecommand> <result> ['drop burger', 'eat burger', 'examine burger', 'examine chair', 'examine staple', 'go east', 'inventory', 'look', 'put burger on chair', 'take staple from chair'] </result>

The admissible commands include 'go east'. According to the game objective, after getting the staple, I need to head east.

4. \*\*Command\*\*: 'go east'

**Now, I'm back in the cellar with the staple. The next instruction is to rest the staple on the shelf inside the cellar. I'll check the description to locate the shelf.**

<description>14 — go west, go west, get staple, go east</description> <result>
-= Cellar =-
You have come into a cellar. Not the cellar you'd expect. No, this is a cellar.
Look over there! a shelf. You shudder, but continue examining the shelf. The shelf is standard. But oh no! there's nothing on this piece of junk. You make a mental note to not get your hopes up the next time you see a shelf in a room. You can see a counter. The counter is normal. But the thing hasn't got anything on it.
There is a closed hatch leading north. You need an unguarded exit? You should try going east. There is an unblocked exit to the west. There is a fondue on the floor.
</result>

The description mentions a shelf in the cellar. I'll put the staple on the shelf.

5. \*\*Command\*\*: 'put staple on shelf'

**Now, I need to check if this completes the game. I'll invoke the \*\*Interface For Obtain Feedback\*\* to see the game's response.**

<feedback>14 — go west, go west, get staple, go east, put staple on shelf</feedback> <result>
You put the staple on the shelf.
Your score has just gone up by one point.
\*\*\* The End \*\*\*
You scored 1 out of a possible 1, in 6 turns.
Would you like to RESTART, RESTORE a saved game, QUIT or UNDO the last command? </result>

The game indicates that I've completed the objective and reached the end. Therefore, the command sequence is successful.
<conclusion>
The command sequence to win the game is: 'go west, go west, get staple, go east, put staple on shelf'
The answer is \boxed{go west, go west, get staple, go east, put staple on shelf}
</conclusion>

# D    LIMITATIONS

In this paper, we propose SITUATEDTHINKER to ground LLM reasoning with various external worlds. Although SITUATEDTHINKER demonstrates noteworthy performance, several avenues for future investigation remain. First, while our experiments and analysis provide valuable insights,

they are presently confined to the textual domain of LLMs and benchmarks. This contrasts with real-world information, which inherently spans multiple modalities, including images, audio, and video. Consequently, investigating methods to effectively incorporate multimodal information into the reasoning process for broader *situated thinking* remains a significant and intriguing challenge for future research. Furthermore, the current scope of our experimentation and analysis is limited exclusively to the English language. Therefore, the applicability and performance of SITUATED-THINKER with interfaces and information presented in languages other than English remains an open question. Addressing this linguistic limitation is crucial for establishing the generalizability of the proposed framework across diverse linguistic contexts. Lastly, the current iteration of our approach primarily addresses deterministic inference problems that possess definitive answers, while largely neglecting open-ended questions or tasks requiring non-deterministic outcomes. While some preliminary exploration within text environments has been conducted, extending the framework to handle complex planning tasks, such as those encountered in interactive environments or robotics, which involve sequential decision-making and managing uncertainty, represents a critical direction for future work and is clearly warranted.

## E    LLM USAGE

In this research, the use of LLMs is confined to the final stages, specifically for refining and proofreading the manuscript. LLMs are employed exclusively to improve the clarity, logical coherence, and linguistic precision of the narrative, ensuring a clear and sophisticated presentation of our ideas. Importantly, LLMs played no role in the foundational components of this study, including the formulation of the research strategy, the design of the experimental framework, or the interpretation of results. We acknowledge full responsibility for the content of the paper.

