# OpenReview forum: "SituatedThinker: Grounding LLM Reasoning with Real-World through Situated Thinking"
_ICLR.cc/2026/Conference — Submitted to ICLR 2026_

### Official Review · Reviewer_pcmm · 2025-10-27

**Soundness:** 3
**Presentation:** 3
**Contribution:** 3
**Rating:** 6
**Confidence:** 3

**Summary:**

This paper introduces SituatedThinner, a novel framework that grounds LLM reasoning in real-world contexts through a paradigm termed "situated thinking." The core idea is to enable LLMs to adaptively interleave internal reasoning steps ("internal actions") with calls to external interfaces ("situated actions") during a single, deliberate reasoning trace. These interfaces provide a unified way to interact with diverse external environments like knowledge bases, code executors, and simulated worlds. The model is trained using a reinforcement learning objective (an adaptation of GRPO) with a simple reward based on final answer correctness, which incentivizes the LLM to autonomously learn when and how to invoke interfaces to acquire information, verify reasoning, and correct errors. Extensive experiments demonstrate significant improvements over strong baselines on multi-hop QA and mathematical reasoning, and, more importantly, show compelling generalization to unseen tasks (KBQA, TableQA) and interfaces (text-based games).

**Strengths:**

**Novel and Well-Motivated Paradigm**: The concept of "situated thinking" is a meaningful and timely contribution. It moves beyond simply injecting external knowledge (as in RAG) towards enabling a dynamic, interactive reasoning process where the model actively queries its environment. This addresses a key limitation of closed-world, parametric reasoning in modern LLMs.

**Impressive Generalization**: The framework's ability to generalize zero-shot to entirely new domains (MedQA, GPQA), tasks (WebQSP, WTQ), and interfaces (TextWorld) is a major strength. This suggests the model learns a generalizable skill of "tool-use" rather than overfitting to specific tasks seen during training.

**Effective and Simple Training Approach**: The use of RL with only a final-answer reward to successfully teach complex behaviors like interface invocation, reflection, and planning is elegant and effective. The ablation studies convincingly show that the model learns these capabilities without explicit supervision on intermediate steps.

**Rigorous Experimental Setup:** The paper provides comprehensive evaluations across a wide range of benchmarks, comparing against a strong set of baselines (including multi-step RAG and other RL-for-reasoning models). The inclusion of training dynamics analysis and qualitative case studies adds depth to the empirical validation.

**Weaknesses:**

**Limited Discussion on Relation to RAG**: While the paper positions itself against "tool-calling" and "search-enhanced" models, a more direct and thorough discussion comparing and contrasting the proposed paradigm with classic and multi-step Retrieval-Augmented Generation (RAG) is lacking. A dedicated subsection explicitly analyzing the fundamental differences (e.g., static knowledge injection vs. dynamic, agentic interaction) would significantly sharpen the contribution and make it more accessible to a broader audience.

**Limited Scope of Evaluation**: The current evaluation is confined to the textual modality and the English language. The limitations section rightly notes this, but it remains a weakness of the current empirical evidence. The framework's performance in multimodal settings or with non-English interfaces remains an open and important question.

**Focus on Deterministic Tasks**: The work primarily addresses tasks with definitive, verifiable answers. Its applicability to more open-ended, creative, or non-deterministic reasoning tasks (e.g., essay writing, complex strategic planning) is not explored and represents a significant boundary for the current method.

**Lack of Assessment on General Capabilities**: The idea of employing a single, powerful LLM for situated reasoning, as opposed to constructing a multi-agent framework, is highly appealing, and I appreciate this elegant approach. However, when training one model to handle such a complex and diverse set of reasoning tasks, its generalization and the preservation of core capabilities become paramount. In this context, the paper currently lacks an assessment of whether the RL fine-tuning causes catastrophic forgetting of the model's general capabilities. It remains unclear if performance on standard benchmarks (e.g., MMLU, BBH) or its general instruction-following ability has degraded after the specialized training for situated thinking. An ablation study comparing the base model and the trained SituatedThinner on such general tasks is crucial to ensure that the acquisition of these novel, specialized skills does not come at the cost of the model's broader competency and linguistic fluency.

It's better to discuss the related works of the agent frameworks for reasoning.

**Questions:**

1) The decision to omit the KL penalty in the GRPO objective is a significant one. To better understand its impact, did you conduct an ablation study on this choice? Specifically, how critical was omitting the KL penalty for achieving the final performance, and what was its effect on the model's behavior and stability during training?

2) Given that the model successfully generalizes to new interfaces, did you observe any cases of "interface confusion" where the model used the wrong interface (e.g., using code execution for a simple fact lookup)? A case study and analysis of such failure modes could be insightful.

---

> ### Author Response · Authors · 2025-11-19
> **Response to Reviewer pcmm**
>
> Dear reviewer pcmm,
>
> Thank you very much for recognizing that our work includes a novel paradigm, an effective training approach, a rigorous experiment setup, and convincing experimental results. Below, we will respond to some of the concerns you have raised.
>
> > Response to W1: Lack of comparison/discussion regarding RAG
>
> Thank you for this suggestion. We have added a dedicated discussion in the Appendix.
>
> - **Classic RAG:** Relies on static knowledge injection. It struggles with multi-hop queries because it cannot dynamically refine the query based on intermediate reasoning.
> - **Situated Thinker:** Dynamic and agentic. It breaks down problems, queries the environment, verifies the result, and iterates within a single thinking process.
>
> > Response to Limited Scope of Evaluation
>
> We agree that multimodal and multilingual evaluations are important next steps. We have acknowledged this in the Limitations section. But since this part of the work is beyond the scope of this paper, we will address this point in our future work.
>
> > Response to W3: Focus on Deterministic Tasks
>
> We focused on deterministic tasks (Math/QA) because they offer verifiable ground truth for reliable evaluation. We agree that open-ended tasks are vital and plan to explore them as evaluation metrics for such tasks mature.
>
> > Response to W4: Potential Catastrophic Forgetting of General Capabilities
>
> This is a critical concern. We conducted an ablation study on general benchmarks (**MMLU** and **GPQA**) to verify this.
> As shown below, *Situated Thinker* actually **improves** performance on these general benchmarks compared to the Base Model. This indicates that our RL framework enhances reasoning fluency without impairing general knowledge.
>
> | **Model** | **MMLU** | **GPQA** |
> | --- | --- | --- |
> | Base Model | 76.89 | 44.44 |
> | **SituatedThinker** | **79.21** | **48.54** |
>
> > Response to Q1: Impact of omitting the KL penalty
>
> Omitting the KL penalty was crucial. The goal of our training is to fundamentally alter the model's behavior (adding tool use/situated thinking), which represents a significant distribution shift from the base model. Enforcing a KL penalty restricts this necessary exploration. As shown in our ablation, re-introducing the KL penalty significantly degrades performance.
>
> |  | **HotpotQA** | **2WikiMultihop QA** | **MusiQue** | **Bamboogle** |
> | --- | --- | --- | --- | --- |
> | **w/ KL penalty** | 0.419 | 0.455 | 0.221 | 0.543 |
> | **SituatedThinker** | 0.433 | 0.464 | 0.235 | 0.552 |
>
> > Response to Q2: Interface Confusion
>
> We analyzed this in Appendix B.2. *SituatedThinker* demonstrates high accuracy in selecting the appropriate interface (e.g., using Code for math, Retrieval for facts), with almost no instances of "interface confusion" affecting the final performance.
>
> ---
>
> Finally, thank you again for your recognition of our work. If you have any other questions, please don't hesitate to let us know.

---

> > ### Comment · Reviewer_pcmm · 2025-11-22
> >
> > Thank you for the author's detailed response, which has addressed my concerns.
> >
> > Due to time constraints, I haven't yet finished reviewing the other reviews and rebuttals, but I will pay attention to the concerns raised by the other reviewers.
> >
> > My current score is already positive, and I will temporarily maintain it for now.

---

### Official Review · Reviewer_c491 · 2025-10-28

**Soundness:** 3
**Presentation:** 3
**Contribution:** 2
**Rating:** 4
**Confidence:** 4

**Summary:**

This paper proposes “Situated Thinker”, which grounds the reasoning process of long chain-of-thought (CoT) LLMs with results returned from external tools. The LLM system is allowed to invoke tools through “interfaces” while generating a long CoT, with results dynamically inserted into the CoT. Two models of 8B and 32B size are trained and evaluated on multi-hop QA, math, and other out-of-distribution reasoning benchmarks

**Strengths:**

- The single-turn formulation of situated thinking paradigm is, to my knowledge, novel. This formulation should greatly simplify the RL training pipeline as compared to more complex multi-turn tool-use enhanced reasoning models.
- The resulting models seem to generalize to unseen domains not explicitly covered in the training data.
- The paper is generally well-written and easy to understand.

**Weaknesses:**

- I find some of the statements about novelty over-claimed:
  - It’s unclear how the claim that situated thinking, which is described as “a new paradigm that allows LLMs to adaptively engage with external environments” in line 91, is different from several LLM-based agents that use retrieval tools, like web-search, e.g., [1,2,3,4], or even baselines ReSearch and Search-R1 (beyond number of tools used).
  - The exact way tool results are incorporated (directly within a single-turn long CoT) may be only differentiator, but I don’t believe this constitutes a fundamentally new paradigm. Rather, as described in Strengths, I consider this a formulation that has some benefits, e.g., simplified training pipelines
- Some issues with baselines:
  - Situated Thinker is trained from Qwen3 family, but the most direct baselines, Search-R1 and ReSearch are trained from Qwen2.5-7B. Given that Qwen3 are generally stronger models, it’s unclear how much the performance difference stems from base model difference vs. methodological improvements. A more reasonable comparison would be to conduct RL training from the same base model if a methodological comparison is the aim, then demonstrating that such approach can improve the performance of better starting models, as is done with Qwen3 experiments.
  - Why do you compare against Qwen3 base LLMs on math benchmarks, and not the post-trained models that are capable of long CoT? Same question for ReAct baselines for OOD tasks?In particular, most current implementations of ReAct agents use instruction-tuned models capable of better instruction following

[1] https://arxiv.org/abs/2509.06283

[2] https://arxiv.org/abs/2507.02592

[3] https://arxiv.org/abs/2507.15061

[4] https://arxiv.org/abs/2508.13167

**Questions:**

- Can you provide more details about how retrieval is done for SituatedThinker? What retriever did you use? Is this retriever the same for RAG-based baselines? How does SituatedThinker's ability depend on the quality of retrievals?

---

> ### Author Response · Authors · 2025-11-19
> **Response to Reviewer c491**
>
> Dear reviewer c491,
>
> We are pleased that you view our method as novel with strong generalization and clear writing. Below, we would like to address your concerns.
>
> > Response to W1: Novelty of "Situated Thinking" vs. existing agents (ReSearch, Search-R1)
>
> The core differentiation is the integration of interaction into the thinking process.
>
> - **Efficiency & Context:** Unlike traditional agentic framework that pause generation to wait for external tools (breaking the context), *Situated Thinker* treats interface interaction as a natural extension of single chain-of-thought. Our method embeds external actions (environmental interactions) and internal thinking into a **single, coherent thought process**. This preserves memory coherence and reduces reasoning overhead.
> - **Generalization:** Unlike ReSearch/Search-R1, which are often optimized for specific tools (web search), we demonstrate generalization across diverse environments (Knowledge Graphs, Python Code, Game Environments).
>
> > Response to W2.1: Fairness of comparison (Qwen3 vs. Qwen2.5)
>
> We agree that a direct retraining of baselines on Qwen3 would be ideal, but the open-source status of the codebase resource constraints prevented this. However, we provide a "Base Model Gap" analysis to contextualize our gains.
>
> - **Base Gap:** Qwen3 outperforms Qwen2.5 by roughly **5-15%** on base reasoning tasks.
> - **Method Gap:** *Situated Thinker* achieves gains of **10-30%** (e.g., Bamboogle: 0.552 vs. 0.432). This significant delta suggests the improvement stems primarily from the methodological advantage of Situated Thinking, not just the base model strength.
>
> |  | **HotpotQA** | **2Wiki** | **MusiQue** | **Bamboogle** |
> | --- | --- | --- | --- | --- |
> | **Qwen2.5-7B** | 0.312 | 0.243 | 0.073 | 0.212 |
> | **Qwen3-8B** | 0.331 | 0.258 | 0.090 | 0.164 |
> | **ReSearch** | 0.406 | 0.447 | 0.217 | 0.432 |
> | **Search-R1** | 0.380 | 0.326 | 0.168 | 0.384 |
> | **SituatedThinker** | 0.433 | 0.464 | 0.235 | 0.552 |
>
> > Response to W2.2: Comparison against Long-CoT post-trained models
>
> We compared against Base LLMs to ensure a fair "apples-to-apples" comparison of the *training framework* itself. The long-CoT model achieves better results by increasing test-time computation. However, since the test-time computation cost cannot be aligned, it is unfair to compare long-CoT models with non-long-CoT models. Additionally, further training on long-CoT requires a large amount of training resources, which is beyond our capabilities.. Additionally, Appendix B.1 demonstrates that *SituatedThinker* still outperforms standard Instruct versions of Qwen3.
>
> | **Model** | **Size** | **Musique** | **Bamboogle** | **WebQSP** | **TextWorld** |
> | --- | --- | --- | --- | --- | --- |
> | **Instruct** | 7B | 0.124 | 0.448 | 34.7 | 22.0 |
> |  | 14B | 0.197 | 0.465 | 58.4 | 46.0 |
> | **SituatedThinker** | **7B** | **0.235** | **0.552** | **66.5** | **42.0** |
> |  | **14B** | **0.255** | **0.512** | **68.7** | **94.0** |
>
> > Response to Q1: Retrieval details
>
> We utilize the **E5-v2** model to generate embeddings and retrieve the top-5 results. This setup is identical to the ReSearch and Search-R1 baselines to ensure fairness. The model's performance is naturally correlated with retrieval quality, but our focus is on how the model *uses* that retrieval to hallucinate less and reason better.
>
> ---
>
> Thank you for your diligent review. We hope our reply can address your questions and concerns, and we are also open to potential other issues.

---

> ### Comment · Reviewer_c491 · 2025-11-25
>
> Thanks to the authors for their response, which has answered some of my questions. I have a few follow-ups
>
> > We agree that a direct retraining of baselines on Qwen3 would be ideal...
>
> I believe a simpler experiment would be to train Qwen2.5 with your approach directly. Even using only Qwen2.5-7B in a controlled experiment would alleviate concerns.
>
> I understand your base model analysis, but I'm hesitant to draw any definitive conclusions about *the impacts of only your training approach*, given there are two confounding factors (base model and training approach). It may very well be the case that Qwen3 can improve more with RL training because it is a stronger model than Qwen2.5. Of course, the opposite may be true; the only want to draw a conclusion on methodological efficacy alone is more controlled experiments.
>
> In my opinion, controlled experimentation is crucial if the main contribution of the work is a method.
>
> > The long-CoT model achieves better results by increasing test-time computation. However, since the test-time computation cost cannot be aligned, it is unfair to compare long-CoT models with non-long-CoT models.
>
> Doesn't SituatedThinker itself use more test-time computation than base models? It calls tools and learns to output longer CoT during RL (Fig 2). A reasonable thing could be to report inference FLOPs vs. performance and analyze Pareto optimality. I ask this specifically for math, as long CoT "thinking" models are the de-facto standard approach to solving math problems currently.
>
> A point of clarification: I'm sympathetic towards limited resources, and am not asking for you to produce a "SituatedThinker-LongCoT"-type model.

---

> > ### Author Response · Authors · 2025-11-27
> >
> > Thank you for your reply and insightful questions. Here is our supplementary response:
> >
> > > I believe a simpler experiment would be to train Qwen2.5 with your approach directly. Even using only Qwen2.5-7B in a controlled experiment would alleviate concerns.
> >
> > We recently re-trained SituatedThinker using Qwen2.5-7B as the backbone. The table below presents a comparison with baseline methods, demonstrating the effectiveness of our method under the same backbone.
> >
> > |  | **HotpotQA** | **2Wiki** | **MusiQue** | **Bamboogle** |
> > | --- | --- | --- | --- | --- |
> > | **Qwen2.5-7B** | 0.312 | 0.243 | 0.073 | 0.212 |
> > | **ReSearch** | 0.406 | 0.447 | 0.217 | 0.432 |
> > | **Search-R1** | 0.380 | 0.326 | 0.168 | 0.384 |
> > | **SituatedThinker w/ Qwen2.5-7B** | **0.428** | **0.454** | **0.229** | **0.486** |
> >
> >
> > > Doesn't SituatedThinker itself use more test-time computation than base models? It calls tools and learns to output longer CoT during RL (Fig 2)
> >
> > The increase in output tokens for SituatedThinker stems primarily from interface feedback. The interface feedback is usually classified as an external computational cost, distinct from the typical test-time computation of LLMs. Crucially, the computational overhead for interface feedback is typically lower than the decoding cost of large-scale LLMs.
> >
> > > A reasonable thing could be to report inference FLOPs vs. performance and analyze Pareto optimality. I ask this specifically for math, as long CoT "thinking" models are the de-facto standard approach to solving math problems currently.
> >
> > We evaluate the performance and inference overhead of both Qwen3-thinking and SituatedThinker on math benchmarks. The results, which account for the token cost of interface feedback for convenience, are presented in the table below. While Qwen3-Thinking achieves higher absolute performance, it is less efficient in terms of *performance per token* (under our hardware conditions, hundreds of tokens can be decoded per second). Consequently, SituatedThinker offers a **superior trade-off between performance and computational cost**.
> >
> >
> > We acknowledge that for complex reasoning problems such as those in mathematics, strong reasoning models based on long-CoT are becoming mainstream. However, the issue of excessive reasoning overhead they present is also attracting attention. Meanwhile, tool-based methods like SituatedThinker are considered to potentially have higher token efficiency [1].
> >
> > | |**Math Performance** |**\#Tokens** |**Performance / \#Tokens (10^-2)** | **Inference Time (s;per data sample)** |
> > | --- | --- | --- | --- | --- |
> > | **Qwen3-Base** | 26.3 | 735.29 | 3.57 | 3.51 |
> > | **SituatedThinker** | 44.8 | 1173.52 | 3.81 | 4.09 |
> > | **Qwen3-Thinking** | 76.5 | 12743.14 | 0.60 | 56.11 |
> >
> > It is also worth noting that on QA benchmarks requiring less complex reasoning, Long-CoT-based models exhibit diminishing returns, or even slight performance degradation. For instance, SituatedThinker achieves a higher average score (0.421) compared to retrieval-augmented Qwen3-Thinking (0.398) across four QA benchmarks. This empirical evidence validates our initial motivation (specifically C1, Lines 063–067), highlighting the distinct advantage of SituatedThinker on knowledge-intensive tasks that heavily rely on external information.
> >
> > [1] Zhao, Yufeng, et al. Dissecting Tool-Integrated Reasoning: An Empirical Study and Analysis. *arXiv preprint* 2025.

---

### Official Review · Reviewer_z4VY · 2025-10-31

**Soundness:** 2
**Presentation:** 2
**Contribution:** 2
**Rating:** 4
**Confidence:** 4

**Summary:**

This paper proposes the situated thinker, an RL-trained framework that enables large language model chain-of-thought reasoning with predefined tool calls (external interfaces), including retrieval, code execution, and other APIs. The authors use GRPO-style RL training to inject tool call capabilities into smaller language models. Trained on MusiQue and Big-Math, the model reports strong gains on mathematical reasoning datasets (e.g., AIME24/25 and MATH500).

**Strengths:**

- This paper is well-motivated and well-written. It’s good to show how general tool-use capabilities can be injected into smaller language models via RL to boost reasoning.
- The results on MATH and knowledge-based reasoning datasets seem strong.

**Weaknesses:**

- Interface design: he authors customized a new interface design for external tool calls. The specifications look highly similar to the Model Context Protocol (MCP). Why not just use MCP, since it’s more universal and enables a wider range of tool use?

- It’s hard to see what actually works. From Table 8, we can see that the tool calls hardly affect model performance on one of the major claimed domains—mathematical reasoning—while on knowledge-based tasks, it’s unclear how retrieval-only methods contribute to the performance. For example, it’s not clear how state-of-the-art retrieval or search-based methods contribute—row 3 in Table 6 might simply show that the proposed situated thinker interface is not effective for retrieval-based tasks.

- Novelty and broader implications: Marrying language models with tool use is not novel and is already a widely recognized direction. I wonder what the broader implications of this project are.

**Questions:**

For TextWorld, how much depends on interface instruction quality vs learned policy?

---

> ### Author Response · Authors · 2025-11-19
> **Response to Reviewer z4VY**
>
> Dear reviewer z4VY,
>
> We are pleased that you find our work well-written with convincing experimental results. Below, we respond to your questions and concerns.
>
> > Response to W1: Interface design vs. Model Context Protocol (MCP)
>
> We chose a custom, simplified interface definition for research purposes to ensure stability and easier ablation studies, whereas MCP is designed for high-complexity production environments.
> However, to demonstrate that our method is not format-dependent, we conducted an experiment replacing our interface definition with an **MCP-style format**. As shown below, *Situated Thinker* maintains robust performance, proving it is compatible with universal standards like MCP.
>
> | **Model** | **Interface** | **Bamboogle** | **AIME24** | **TextWorld** |
> | --- | --- | --- | --- | --- |
> | **SituatedThinker** | Original | 0.552 | 27.0 | 21/50 |
> |  | **MCP-style** | **0.547** | **26.5** | **23/50** |
>
> > Response to W2: Clarifying the contribution of retrieval vs. internal reasoning (Tables 6 & 8)
>
> We apologize if the interpretation of the tables was unclear.
> - **Robustness (Table 8):** The slight changes in Table 8 are intended to show **robustness against redundant interfaces**. Even when provided with *all* interfaces (some irrelevant), the model selects the correct one and maintains performance.
> - **Effectiveness (Table 6):** Row 1 shows performance *without* any interfaces, which results in significant degradation. This confirms the necessity of the interface. Rows 2 and 3 represent ablations (Retrieval-only vs. Code-only). As expected, **Retrieval** dominates knowledge tasks (Bamboogle), while **Code** dominates math tasks (AIME24).
> - **Retrieval Quality:** We used a standard retriever (E5-v2) across all experiments, which aligns with baselines (Search-R1 & ReSearch). While a better retriever would likely improve results further, our contribution focuses on the *reasoning paradigm* rather than the retrieval subsystem.
>
> > Response to W3: Novelty and broader implications
>
> While tool use is a known direction, our contribution is the shift from a "Plan-and-Act" (multi-turn) paradigm to a "Situated Thinking" (single-turn) paradigm.
>
> - **Cognitive Coherence:** By integrating external interaction into a single thought process, we mitigate the context dilution and error accumulation inherent in multi-turn agents (which cause 20-30% degradation [1][2]).
> - **Open-Source DeepResearch:** We believe this work serves as a foundational step toward open-source frameworks similar to OpenAI's DeepResearch, moving beyond simple search-augmented generation toward autonomous, self-correcting inquiry.
>
> > Response to Q1: TextWorld performance: Instruction quality vs. Learned Policy
>
> The performance is primarily driven by the **learned policy**. As shown in Table 3, untrained Base LLMs score only 8.0 on TextWorld even with instructions. *Situated Thinker*, through RL, learns *how* to invoke the interface and integrate feedback, achieving a score of 42.0 (7B) and 94.0 (14B).
>
> Table3:
>
> | **Model** | **Size** | **Interfaces** | **MedQA** | **GPQA** | **WebQSP** | **WTQ** | **TextWorld** |
> | --- | --- | --- | --- | --- | --- | --- | --- |
> | **Vanilla LLMs** | **7B** | **✗** | 3.9 | 6.1 | 41.7 | 6.5 | 10.0 |
> |  | **14B** | **✗** | 69.6 | 39.4 | 53.1 | 10.5 | 24.0 |
> | **ReAct** | **7B** | **✓** | 5.7 | 5.6 | 22.1 | 35.7 | 8.0 |
> |  | **14B** | **✓** | 71.9 | 35.9 | 56.3 | 41.6 | 28.0 |
> | **ReSearch** | **7B** | **✓** | 43.6 | 20.2 | 25.2 | 8.7 | 4.0 |
> | **Search-R1** | **7B** | **✓** | 21.8 | 15.2 | 15.9 | 5.1 | 2.0 |
> | **SituatedThinker** | **7B** | **✓** | 58.1 | 25.8 | 66.5 | 68.9 | 42.0 |
> |  | **14B** | **✓** | 77.9 | 53.0 | 68.7 | 69.7 | 94.0 |
>
> Table6:
>
> | **Training Configuration** | **Bamboogle** | **AIME24** | **TextWorld** |
> | --- | --- | --- | --- |
> | **w/o interfaces** | 0.208 | 11.3 | 14.0 |
> | **w/ retrieval only** | 0.440 | 22.4 | 40.0 |
> | **w/ code only** | 0.336 | 24.7 | 36.0 |
> | **SituatedThinker** | 0.552 | 27.0 | 42.0 |
>
> [1] Liu, Qiang, et al. Multi-turn Training with Basic Human Feedback Helps Little on LLM Reasoning. *arXiv preprint 2025*.
>
> [2] Li, Yubo, et al. Beyond single-turn: A survey on multi-turn interactions with large language models. *arXiv preprint 2025*.
>
> ---
>
> Finally, thank you again for your efforts and time. We hope our response can address your questions. If you have any other questions, please let us know.

---

### Official Review · Reviewer_v1GG · 2025-10-31

**Soundness:** 3
**Presentation:** 2
**Contribution:** 2
**Rating:** 4
**Confidence:** 4

**Summary:**

This paper proposes a framework, SituatedThinker, which aims to ground the reasoning of LLMs in the real world. The core of this method is to enable LLMs to learn to adaptively combine their internal knowledge with external information obtained through predefined "Interfaces". The authors use RL to train the LLM and achieve a good performance in multiple tasks.

**Strengths:**

1. The problem of grounding the LLM in real-world situations is important to explore.

2. On multi-hop question-answering and mathematical reasoning, the model significantly outperforms baseline methods.

**Weaknesses:**

1. I am still unclear about the distinction between "Situated Thinking" and standard tool learning. The "Interfaces" defined in the paper, which include the description, the specified input format, and outputs, do not seem fundamentally different from the LLM agent function calling used in the community today. The only difference is the representation, such as text-based tags versus JSON Schema, which makes the contribution incremental.

2. One of the contributions is that the paper claims that the model can generalize to unseen interfaces. However, this does not seem particularly compelling. Most current instruction-tuned models already possess a "zero-shot tool use" capability: as long as a clear description and usage format for a new tool (e.g., the format for OpenAI's function calling) is provided in the message, the model can follow these instructions to use it. This may weaken the contribution as well.

**Questions:**

1. Would performing SFT first to equip the model with better format-following capabilities make the RL training smoother? Why was this setting not considered?

2. There is a spelling error in the Interface Template in Section 2.1.

---

> ### Author Response · Authors · 2025-11-19
> **Response to Reviewer v1GG**
>
> Dear reviewer v1GG,
>
> We are very pleased that you recognize the value of our questions and that our model demonstrates convincing performance. Below is our response to your concerns and doubts.
>
> > Response to W1: Distinction between "Situated Thinking" and standard tool learning/function calling
>
> While the "Interfaces" may superficially resemble standard function calling (e.g., JSON Schema), the fundamental distinction lies in the cognitive architecture of the interaction.
>
> - **Standard Frameworks (e.g., ReAct, Toolformer):** These rely on a fragmented "Plan-and-Act" cycle. The model generates a thought, stops to call a tool, waits for the environment, and then starts a new generation turn. This multi-turn fragmentation often leads to context dilution and error accumulation.
> - **Situated Thinking:** Our method embeds external actions (environmental interactions) and internal thinking into a **single, coherent thought process**. This seamlessly intertwines reasoning with action, maintaining the "thread of thought."
> - **Performance Impact:** As cited in our paper (and recent studies like [1][2]), multi-turn interactions can degrade performance by 20-30% due to context loss. Our single-turn approach maintains reasoning coherence, reduces latency, and optimizes token efficiency.
>
> > Response to W2: Generalization to unseen interfaces and "Zero-Shot" capabilities
>
> Instruction-tuned models possess zero-shot tool capabilities. However, our experiments demonstrate that this capability is often brittle and insufficient for complex reasoning.
>
> - **Inadequacy of Current Models:** As noted in Appendix B.1, standard instruction-tuned LLMs often struggle to process external environmental feedback effectively within a single thinking process [3].
> - **Empirical Evidence:** We compared *Situated Thinker* against instruction-tuned models (e.g., Qwen3-Instruct) in a zero-shot setting. As shown in the table below (from Appendix B.1), *Situated Thinker* significantly outperforms the instruction-tuned baseline, particularly on out-of-domain tasks like **WebQSP** (66.5 vs. 34.7) and **TextWorld** (42.0 vs. 22.0).
>
> | **Model** | **Size** | **Musique** | **Bamboogle** | **WebQSP** | **TextWorld** |
> | --- | --- | --- | --- | --- | --- |
> | **Instruct** | 7B | 0.124 | 0.448 | 34.7 | 22.0 |
> |  | 14B | 0.197 | 0.465 | 58.4 | 46.0 |
> | **SituatedThinker** | **7B** | **0.235** | **0.552** | **66.5** | **42.0** |
> |  | **14B** | **0.255** | **0.512** | **68.7** | **94.0** |
>
> > Response to Q1: Why not use SFT (Supervised Fine-Tuning) for a "cold start"?
>
> We deliberately chose a Zero-RL setting for two key reasons:
>
> - **Avoid Format Overfitting/Leakage:** SFT can bias the model toward specific training data patterns (leakage) or restrict its exploration capabilities. We wanted to assess the model's ability to *learn* the interaction policy purely through reinforcement.
> - **Efficiency:** Our experiments show that base LLMs quickly learn the correct format in the early stages of RL training without needing an SFT warm-up phase. This eliminates the costly requirement of constructing high-quality SFT trajectories.
>
> > Response to Q2: Spelling error in Section 2.1
>
> Thank you for spotting this. We have corrected the spelling error in the Interface Template in the revised paper.
>
> [1] Liu, Qiang, et al. Multi-turn Training with Basic Human Feedback Helps Little on LLM Reasoning. *arXiv preprint 2025*.
>
> [2] Li, Yubo, et al. Beyond single-turn: A survey on multi-turn interactions with large language models. *arXiv preprint 2025*.
>
> [3] Laban, Philippe, et al. Llms get lost in multi-turn conversation. *arXiv preprint 2025*.
>
> ---
>
> Finally, thank you again for your thoughtful review. We hope this response addresses your concerns, and we welcome further discussion.

---

### Author Response · Authors · 2025-11-19
**General Response**

We sincerely thank all reviewers for their thorough and thoughtful reviews of our paper. We are grateful for the positive feedback on the novelty of the "Situated Thinking" paradigm and the strong experimental results across diverse domains. The constructive suggestions provided have helped us significantly improve the clarity and quality of our work. We have provided a detailed response to each reviewer. We have revised the paper according to the reviews, and the edits have been highlighted in BLUE. Here we highlight our major revisions and general clarifications regarding the core contributions and experimental settings. We hope our responses can properly address your concerns.

- **Clarification of "Situated Thinking" Paradigm vs. Standard Agents** (Reviewers v1GG, z4VY, c491): We want to clarify the fundamental distinction between our method and standard "Plan-and-Act" frameworks (e.g., ReAct, Toolformer). While they share surface-level similarities in tool usage, our approach integrates external interaction into a **single, coherent thought process**. We have added discussions to emphasize that this single-turn paradigm mitigates the context dilution and error accumulation inherent in multi-turn interactions (which can cause 20-30% performance degradation). We further differentiate our work from commercial baselines (like ReSearch) by demonstrating generalization across diverse environments beyond just web search.
- **Robustness to Interfaces and General Capabilities** (Reviewers z4VY, pcmm): We provide additional analyses to demonstrate the robustness of our model. **Regarding Interface Formats:** We conducted experiments using **MCP-style** interfaces (see Response to z4VY), showing that *Situated Thinker* maintains robust performance regardless of the specific interface definition. **Regarding General Knowledge:** We included ablation studies on **MMLU** and **GPQA** (see Response to pcmm) to confirm that our RL training framework enhances reasoning fluency without suffering from catastrophic forgetting of general capabilities; in fact, it yields slight improvements over the base model.
- **Fairness of Baselines and Training Setup** (Reviewers c491, v1GG, pcmm): We have clarified our experimental choices to ensure fair comparisons. We explicitly analyzed the **Base Model Gap** between Qwen3 and Qwen2.5 to contextualize our performance gains, showing that the improvements stem primarily from our method rather than the base model's strength. Furthermore, we justified our **Zero-RL setting** (without SFT warm-up) and the omission of the **KL penalty**. Our analysis confirms that these settings are essential for allowing the model to learn the interaction policy effectively and explore necessary distribution shifts without being restricted by the base model's priors.

---

### Meta-Review · Area_Chair_rFdz · 2026-01-06

**Summary:**

While the authors have addressed several technical validity concerns during the rebuttal—specifically regarding baseline fairness (retraining on Qwen2.5) and catastrophic forgetting—the submission ultimately falls short of the novelty threshold required for acceptance at ICLR. The central claim of the paper is the introduction of "Situated Thinking" as a "novel paradigm". However, the core concept of interleaving tool use, retrieval, and reasoning within a chain-of-thought (often referred to as "Interactive RAG," "Agentic RAG," or "Tool-Integrated Reasoning") has become a densely populated and established research area by late 2025.

The paper positions its "single-turn" architecture as a breakthrough differentiator from multi-turn agents like ReAct. While this offers an efficiency optimization, it represents an incremental engineering refinement rather than a fundamental paradigm shift. Similar "deep integration" of tools into the generation stream has been explored in concurrent works (e.g., ToolGen, SimpleTIR, and various "Thinking" models). The logic that this constitutes a distinct "paradigm" is overstated.

Furthermore, while the "Zero-RL" training strategy is effective, applying reinforcement learning to optimize tool-use policies is a known methodology. The efficiency gains shown in the rebuttal (Pareto optimality) are valuable but do not compensate for the lack of conceptual novelty. The paper is well-executed but ultimately incremental, refining existing "Agentic RAG" workflows rather than introducing a transformative new direction.

**Reviewer Concerns:**

1. Baseline Fairness & Backbone Discrepancy (Resolved): Reviewer c491 noted that the original submission compared SituatedThinker (trained on the stronger Qwen3) against baselines like ReSearch and Search-R1 (trained on the weaker Qwen2.5), making it unclear if gains were due to the method or the base model. The authors conducted a strictly controlled experiment by retraining SituatedThinker on the Qwen2.5-7B backbone. The method still outperformed baselines (e.g., 0.428 vs. 0.406 on HotpotQA), proving the improvements are methodological.

2. Novelty of the "Situated Thinking" Paradigm (Clarified): Multiple reviewers (v1GG, z4VY, c491) questioned how this differs from standard tool-use agents like ReAct or existing function-calling frameworks. The authors clarified that their approach uses a single-turn cognitive architecture where tool use is embedded in the chain-of-thought, contrasting it with the fragmented, multi-turn "Plan-and-Act" cycles of agents like ReAct. They argued this mitigates the context dilution and error accumulation (20-30% degradation) seen in multi-turn frameworks. However, I still think similar interactive tool use and RAG is mainstream nowdays and is not a big innovation.

**Reviewer Scores:**

The second reviewer might increase the score. Other reviewers' main focus was on the novelty. Because the similar paradigm to interactive tool use and agentic RAG, I don't think their novelty concerns are solved.

---

### Decision · Program_Chairs · 2026-01-26

Reject